# Involvement of the Opioid Peptide Family in Cancer Progression

**DOI:** 10.3390/biomedicines11071993

**Published:** 2023-07-14

**Authors:** Manuel Lisardo Sánchez, Francisco D. Rodríguez, Rafael Coveñas

**Affiliations:** 1Laboratory of Neuroanatomy of the Peptidergic Systems, Institute of Neurosciences of Castilla and León (INCYL), University of Salamanca, 37007 Salamanca, Spain; lisardosanchez8@gmail.com; 2Department of Biochemistry and Molecular Biology, Faculty of Chemical Sciences, University of Salamanca, 37007 Salamanca, Spain; lario@usal.es; 3Group GIR-USAL: BMD (Bases Moleculares del Desarrollo), University of Salamanca, 37007 Salamanca, Spain

**Keywords:** enkephalin, endorphin, dynorphin, opioid receptor, apoptosis, metastasis, cancer progression, tumor cell, angiogenesis

## Abstract

Peptides mediate cancer progression favoring the mitogenesis, migration, and invasion of tumor cells, promoting metastasis and anti-apoptotic mechanisms, and facilitating angiogenesis/lymphangiogenesis. Tumor cells overexpress peptide receptors, crucial targets for developing specific treatments against cancer cells using peptide receptor antagonists and promoting apoptosis in tumor cells. Opioids exert an antitumoral effect, whereas others promote tumor growth and metastasis. This review updates the findings regarding the involvement of opioid peptides (enkephalins, endorphins, and dynorphins) in cancer development. Anticancer therapeutic strategies targeting the opioid peptidergic system and the main research lines to be developed regarding the topic reviewed are suggested. There is much to investigate about opioid peptides and cancer: basic information is scarce, incomplete, or absent in many tumors. This knowledge is crucial since promising anticancer strategies could be developed alone or in combination therapies with chemotherapy/radiotherapy.

## 1. Introduction

New molecular targets and compounds that specifically destroy tumor cells must be investigated to reach higher cure rates and a better quality of life for cancer patients. The peptidergic systems are one of the promising targets that open up new research lines and possibilities to explore new antitumor therapies and improve cancer diagnosis. These systems have attracted increasing interest because many peptides mediate cancer progression [1,2,3,4,5,6,7]. Peptides, through autocrine, paracrine, and endocrine mechanisms, favor the mitogenesis, migration, and invasion of tumor cells, promoting metastasis; exert an anti-apoptotic mechanism in these cells, and facilitate angiogenesis/lymphangiogenesis [1,5,6,7]. Tumor cells overexpress peptide receptors, which represent crucial targets for developing specific treatments against cancer cells using peptide receptor antagonists, which promote apoptosis in tumor cells, block the migration of cancer cells, and inhibit angiogenesis [1,5,6,7]. In combination therapy with chemotherapy, these antagonists exert a synergic effect and decrease the side effects promoted by cytostatics [8]. A common specific anticancer strategy using peptide receptor antagonists seems possible, irrespective of the tumor type [1,5,6,7]. Moreover, some peptide receptors are involved in the viability of tumor cells [9]. The overexpression of the peptidergic systems has been associated with tumor size, higher tumor aggressiveness, increased relapse risk, worse sensitivity to chemotherapy agents, and poor prognosis [1,5,6,7].

The involvement of opioid peptides in cancer has been widely demonstrated [10,11,12,13]; some studies have shown that opioids exert an antitumoral effect, whereas others have shown that opioid peptides promote tumor growth, metastasis, and vascularization [14,15,16,17,18]. Accordingly, this review aims to update the findings regarding the involvement of opioid peptides (enkephalins, endorphins, and dynorphins) in cancer development (e.g., mitogenesis, metastasis, angiogenesis); to suggest anticancer therapeutic strategies targeting the opioid peptidergic system (e.g., using opioid-receptor antagonists) and to highlight the main research lines to be developed in the future focused on the involvement of enkephalins, endorphins, and dynorphins in tumor development and progression.

## 2. The Endogenous Opioid System: Peptides and Receptors

Three genes, namely *PDYN*, *PENK*, and *POMC*, encode multiple endogenous peptides forming three distinct families of classical endogenous opioid peptides. A fourth gene, *PNOC*, encodes a group of non-classical opioid peptides more recently discovered to be closely related to opioid physiological action [19,20] (Figure 1). Endogenous opioid peptides may be classified into different families according to their preferential affinity for the four opioid receptors (OR), the so-called classical μ, δ, κ (MOP, DOP, KOP), and the opioid-related nociceptin/orphanin (NOP), respectively, into endomorphins, β-endorphins, enkephalins, dynorphins, and nociceptins [19,21]. A fifth opioid-like receptor type with a role in cell proliferation, activated by methionine-enkephalin (MET), acts at the nucleus membrane and is named the opioid growth factor receptor, or receptor zeta (ζ) (OGP/ZOP) [19,22,23]. The receptor has no sequence homology or structural resemblance with the opioid receptors, but naltrexone antagonizes methionine-enkephalin’s specific binding and action on ZOP [22].

Endogenous opioids participate in numerous actions in the central nervous system and peripheral tissues, modulating analgesia, stress, memory, cardiovascular and respiratory control, gland secretion, development, and angiogenesis (reviewed in [24,25]).

### 2.1. Endomorphins

Endogenous tetrapeptides, endomorphins 1 (H_2_N-YPWF-amide) and 2 (H_2_N-YPFF-amide) (Figure 2) were isolated from the mammalian brain. These tetrapeptides exhibit high affinity and selectivity for the μ opioid receptor and may be considered the natural ligands of MOP [26]. The origin of endomorphins has not been determined [27]. Mexneurin, a 91-amino-acids protein encoded by gene *Trnp1* (mouse) (Uniprot code I6L9U2, [19]), has been proposed as a possible precursor from which the peptides may originate after posttranslational proteolysis and amidation of the C-terminal Phe residue [28].

### 2.2. β-Endorphins

The human *POMC* gene has the 2p23.3 chromosomal localization. It encodes the preprotein proopiomelanocortin that bears tissue-specific posttranslational proteolysis catalyzed by prohormone convertases PC1 and PC2 [30]. The preprotein may yield ten different bioactive peptides, including adrenocorticotropin (ACTH), lipotropin, melanotropins, and β-endorphins (Figure 1) [19,31]. Although multiple forms of β-endorphin exist, the most common form has 31 amino acids (Figure 2). β-endorphins are preferential endogenous agonists of μ opioid receptors (MOR) but bind and activate other opioid receptors (DOP and KOP). They display multiple effects, including pain relief, stress control, brain metabolism homeostasis, eating and drinking behavior, and memory modulation, to name a few [24,32,33].

### 2.3. Enkephalins

The *PENK* gene, located at chromosome 8q12.1, encodes a preprotein, proenkephalin A, posttranslationally processed by proteolysis to give several peptide products, including mainly the pentapeptides leucine-enkephalin (LEU) and MET [19,31]. Both peptides may also originate from *PDYN* and *POMC* genes (Figure 1).

Soon after demonstrating the opioid receptor sites in nervous tissue [34], two pentapeptides, LEU and MET, were first isolated from the pig brain and found to inhibit in vitro electrically stimulated smooth muscle contraction of isolated guinea pig ileum preparations. Their effect was found to be blocked by the opioid antagonist naloxone [35]. The pentapeptides share the N-terminal sequence YGGF with β-endorphin and dynorphin A (1–17) (Figure 2). Both enkephalins and endorphins bind to μ and δ receptors with similar affinity, even though enkephalins are considered delta opioid ligands [36]. MET is also considered an inhibitory growth regulator. It specifically binds to OGR/ZOR, a protein at the nuclear membrane. After binding to MET, it penetrates the nucleus interior, affecting cell proliferation [22,37,38]. However, many aspects of this receptor’s function and three-dimensional structure remain unsolved and further studies are needed to establish its role.

### 2.4. Dynorphins

The human *PDYN* gene, with 20p13 chromosomal localization, encodes a preprotein, prodynorphin, from which different bioactive peptides with preferential selectivity for KOP originate after posttranslational proteolysis, including β-neoendorphin, dynorphin, LEU, and leumorphin (Figure 1) [19]. A representative of this family of peptides, the octapeptide dynorphin A (1–8), is considered a natural ligand of KOP. However, it also activates MOP, DOP, and NMDA (N-methyl-D-aspartate) receptors [39,40]. Dynorphin A (1–8) has a role in the modulation of analgesia, stress, emotional states, and memory, to mention a few [24]. It also has a role in neuroprotection by inhibiting apoptosis and oxidative stress mechanisms [40].

### 2.5. Nociceptins

The human *PNOC* gene, localized at chromosome 8p21.1, encodes prepronociceptin, which after proteolytic processing, generates several peptides, including the triacontapeptide nocistatin and two heptadecapeptides, nociceptin and orphanin FQ2 (Figure 1 and Figure 2) [19]. The representative peptide of this family, nociceptin, binds to the protein opioid-related receptor NOP [23]. Nociceptin was first isolated and found to induce a decrease of latencies in hot plate and tail flick tests after intracerebroventricular injection in mice [41]. Later, it was discovered that nociceptin mediated the blocking of opioid-receptors-induced analgesia [42]. The nociceptins family have a Phe residue in the N-terminus, compared with the N-terminal Tyr in the other families of endogenous peptides (Figure 2), and participate in numerous biological activities, such as substance abuse, analgesia control, memory, posttraumatic stress disorder, neuronal differentiation, and cell proliferation [43,44,45].

### 2.6. The Structure and Dynamics of the Opioid Receptors (OP)

Opioid receptors MOP, DOP, KOP, and NOP belong to the GPCR (G-protein-coupled receptors) family of proteins. The ζ Receptor (ZOP) does not belong to the GPCR family of proteins, and its 3D architecture remains undefined.

Two principal methods, X-ray diffraction and cryo-electron microscopy, are potent methods to define receptor conformations that facilitate structure-based drug design (SBDD) [46]. In this study, we highlight these proteins’ structural and functional features and their impact on intracellular signaling pathways related to possible disturbances of cell proliferation and cell life control leading to cancer.

The resolution of the architecture and ligand binding pockets of the OR fits well with the pharmacological concept of “message address,” which establishes that ligands have a module responsible for the message (recognition and efficacy) and another governing the address (additional recognition and selectivity) when contacting their binding site [47]. Together with structural analysis, it is paramount to use in vivo models to grasp their physiological role. In this respect, the current availability of mice knock-out models is valuable [48].

#### 2.6.1. The μ Receptor (MOP)

The human MOP is encoded by the gene *OPRM1* (chromosomal location 6q25.2). It has 400 amino acids. Posttranslational processing events of MOP include a disulfide bridge and N-glycosylation, phosphorylation, and lipidation sites (Figure 3A) [49,50].

The existence of multiple spliced and truncated variants of the receptor with functional significance and tissue-specific expression has added complexity and plasticity to the signaling panorama and intracellular events triggered by MOP [55,56].

The first X-ray diffraction study on the three-dimensional structure of murine MOP covalently bound to the opioid antagonist β-FNA (beta-funaltrexamine) through K233^5.39^ (Ballesteros and Weinstein numbering for CPCRs [57]) revealed the existence of a ligand pocket significantly exposed to the extracellular space and defined by both hydrophobic and polar interactions with amino acid residues positioned on transmembrane helices 3, 6, and 7. The exposure of opioid ligands to the receptor’s extracellular surface may explain the rapid half-lives of dissociation constants of opioid drugs binding to this receptor [58]. Huang et al. [59] determined the active state structure of murine MOP (stabilized with a G-protein mimetic nanobody) bound to the potent morphinan agonist BU72 (a bridged pyrrolidine morphinan). The binding pocket for BU72 is similar to the one described for β-FNA and secures its position within the orthostatic pocket through hydrophobic, aromatic, and polar interactions.

Several analyses have provided relevant information on the structural dynamics of MOP. Recent cryo-electron microscopy studies on MOP architecture have unveiled valuable information regarding the functional states of the receptor leading to the broad plasticity and pliability of MOP signaling. Zhuang et al. [52] determined the structure of the human MOP bound to several agonists, including morphine (Figure 3B), fentanyl, PZM2, SR17018, and TRV130. They applied mutagenesis analysis of the transmembrane domain 6 and 7 interfaces to define the binding pocket occupied by the agonists analyzed. Interestingly, morphine and fentanyl bind to a pocket delimited by transmembrane region 3 (TM3) and the interface between TM6/7. In contrast, other agonists, including SR17018, preferentially bind to TM3 and interact more loosely with the interface TM6/7 (Figure 3C). The consequences of this different coupling mode may have biological significance related to biased signaling through preferential β-arrestin activation over Gi when the binding to the TM6/7 is not tight enough [52].

Cryo-electron microscopy of MOP bound to the synthetic selective peptide agonist DAMGO (H-Tyr-d-Ala-Gly-N(Me)Phe-Gly-OH) and a heterotrimeric Gi protein with not bound nucleotide unveiled a specific pocket where the peptide sits with its N-terminus amino group occupying a similar site compared with BU72, establishing a salt bride with D147^3.32^ and a hydrogen bond with Y326^7.43^. However, the C-terminus continues towards the extracellular loops of MOP [60].

The elucidation of binding sites for agonists and antagonists is paramount to defining receptor activation and inactivation states that may determine biased signaling through Gi or β-arrestin transducers. Docking and molecular dynamics simulation analysis of MOP1 provided relevant clues on the specific interaction of receptor-biased agonist PZM21 that establishes strong interactions with TM residue Y^7.43^ and TM3 residues (D^3.32^ and Y^3.33^), leading to an intracellular opening of the receptor protein that facilitates G-protein binding [61]. Determination of cryo-EM structures of MOP-Gi bound to biased agonists PZM21 and its naphthyl-substituted acryl amide analog derivative, FH210, that exhibits a more pronounced G-protein bias, compared with the parent compound PZM21 [62], revealed the existence of an extended receptor pocket where PZM21 interacts with polar groups D^3.32^ and Y^7.43^, and its phenol group connects with H^6.52^ by a water-mediated interaction. The PZM21 thiophene group sits in a hydrophobic indentation of MOP delimited by I^3.29^, V^3.28^, and Q^2.60^ residues (Figure 4).

The agonist FH210 fits its naphthyl group in the hydrophobic indentation delimited by residues of the transmembrane domains 3 and 2 better than the thiophene group of PZM21 does (Figure 4). This binding mode may explain a stable conformation that favors the receptor activation of Gi protein more successfully than PZM21, avoiding β-arrestin signaling [62]. These observations open new perspectives concerning drug design targeting specific receptor transducers and possible biased intracellular responses [63].

#### 2.6.2. The δ Receptor (DOP.)

The human DOP protein is encoded by gene *OPRD1* (chromosomal location 1p35.3). Its primary structure consists of 372 amino acids. Posttranslational processing events of DOP include a disulfide bridge and N-glycosylation, phosphorylation, and lipidation sites (Figure 5A) (GPCR database).

Two types of DOP have been pharmacologically defined: type 1, activated by DPDPE [(D-Pen 2, D-PenS)enkephalin] and antagonized by BNTX (7-benzylidene naltrexone), and type 2, sensitive to DELT II [(D-Ala2) deltorphin II], antagonized by Naltribe, (a benzofuran derivative of naltrindole) [68]. The functioning of DOP receptor types may have physiological implications, as a coordinated role in pain and anxiety modulation has been reported [69].

X-ray diffraction analysis of murine DOP bound to antagonist naltrindole reported the hole where the ligand establishes contacts with amino acid residues situated in transmembrane domains 3, 6, and 7 (for example, D^3.32^, Y^3.33^, I^6.51^, H^6.52^, V^6.55^, W^6.58^, L^7.35^, Y^7.43^) (Figure 5B) [64]. This space accommodates the structure of naltrindole. Leucine in position 300^7.33^ keeps contact with the naltrindole indole ring and determines naltrindole selectivity. Interestingly, positions W^7.35^ in MOP and Y^7.35^ in KOP offer a significant steric hindrance hampering stable connection with the naltrindole indole group [64].

The resolution of human DOP in its inactivated state bound to naltrindole [65] provided a similar structure to mouse DOP. It contributed new information regarding the molecular bases for receptor activation, including the definition of the allosteric sodium site, acting as a molecular switch (negative cooperativity), situated amid a plexus of polar interactions, in close contact with neighboring residues S^3.39^, N^3.35^, N^7.45^, and D^2.50^ [65] (Figure 5C).

Claff et al. (2019) reported an analysis of the structure of activated DOP bound to the agonist DPI-287 (4-[(4-benzyl-2,5-dimethylpiperazin-1-yl)-(3-hydroxyphenyl)methyl]-N, N-diethyl benzamide) [66]. The authors contributed structural details explaining agonist binding, selectivity, MOP activation, and the role of the allosteric sodium site, which appears disintegrated upon agonist binding. DPI-287 sits on a cavity of DOP where certain amino acid positions establish close contacts (for example, T^2.56^, Q^2.60^, D^3.32^, Y^3.33^, Y^7.43^) (Figure 5D).

A recent structural dynamics and pharmacological analysis on human DOP bound to peptide deltorphin II (Y[D-Ala^2^]FEVVG [67] establishes the accommodation of the peptide within the receptor protein and indicates differences that explain distinct selectivities of ligands. The deltorphin N-terminus region Y[D-Ala^2^]F sits at the bottom of the orthosteric cavity, whereas the C-terminus end, EVVG, interacts with the extracellular regions of TM2, TM6, TM7, and ECL3. The contact of deltorphin glutamic acid in position 4 through a salt bridge with K214^5.39^ and a hydrogen bond with Y129^3.33^ determines binding selectivity. Additionally, valines in positions 5 and 6 adapted to a hydrophobic pocket defined by V281^6.55^, W284^6.58^, L300^7.35^, H301^7.36^, and I304^7.39^. Additionally, the carbonyl group of valine 5 of the peptide makes an ionic contact with R29^ECL3^, which forces an inward movement of TM6, TM7, and ECL3 toward the orthostatic binding site that contributes to ligand selectivity (Figure 5E) and differs from the one established by agonist DPI-287 with DOP [66].

Unveiling the atomic interactions of DOP with ligands and determining interactome maps [70] are excellent tools to speed drug design to obtain new compounds with defined properties to fine-modulate receptor activity.

#### 2.6.3. The κ Receptor (KOP)

The human KOP is encoded by the gene *OPRK1* (chromosomal location 8q11.23). It has 380 amino acids. Posttranslational processing events of KOP include a disulfide bridge, N-glycosylation, and lipidation sites (Figure 6A) [49,50].

Early NMR structure and dynamics studies on the interaction of dynorphin (1–13) peptide with human KOP [73] shed light on the contacts and conformations adopted by dynorphin when accommodated by the receptor. The peptide N and C-termini adopt flexible and disordered conformations, and the central part forms a helical turn. The observed disordered and flexible conformations of the peptide may reflect a process of binding and activation where ligand and receptor adapt to each other through intermediary-bound states with functional meaning.

The crystal structure of KOP bound to selective antagonist JDTic ((3R)-7-hydroxy-N-[(2S)-1-[(3R,4R)-4-(3-hydroxyphenyl)-3,4-dimethylpiperidin-1-yl]-3-methylbutan-2-yl]-1,2,3,4-tetrahydroisoquinoline-3-carboxamide) helps to understand the fitting of the ligand and its affinity, selectivity, and potency [74]. The ligand interacts with amino acid residues through polar and hydrophobic contacts, delimiting the binding pocket (Figure 6B).

Determination of the crystal structure of KOP bound to non-selective agonist MP1104, stabilized by nanobody Nb39, and comparison of active and inactive states of the receptor gave insights into the conformations explaining the pharmacology and biased signaling of KOP [75]. Additionally, the use of nanobodies to regulate KOP activity has been further developed, and an inactive state of KOP has been resolved with nanobody Nb6 [72].

By contrasting the inactive state ensemble KOP-JDTic [74] with the active complex KOP-MP1104, the authors communicated that in the active state, ECL2 and TM domains 4–6 move inwardly toward the receptor complex. Consequently, the orthosteric cavity contracts and its volume diminishes by around 10% compared with the cavity in the inactive structure, a similar finding observed for MOP [59]. A common feature between both complexes is the interaction with D1383.22 through an ionic interaction and the contact of the phenolic moiety with TM5. The cyclopropyl methyl group of MP1104 sits in a hydrophobic hole built by residues W2876.48, G3197.42, and Y3207.43 (Figure 6C). Mutagenesis analysis of these positions pointed out that mutations G3197.42L and Y3207.43L showed a reduced G protein and β-arrestin activation, whereas, with mutation W2876.48L, a selective reduction of β-arrestin recruitment was observed [75].

In a recent cryo-electron microscopy study, different active states of KOP were elucidated by analyzing complexes of KOP and multiple G-protein heterotrimers, which may help better understand ligand selectivity and receptor–G-protein interaction [76].

Resolved structure of KOP bound to peptide dynorphin (1–13), a few contacts differ from the site occupied by MP1104. The amine group of Y1 at the N-terminus sits at the bottom of the orthosteric binding pocket, similarly to MP1104 contacts through its tertiary amine group (Figure 6C). However, the peptide exhibits additional contacts towards the top region of the binding site, with R6 and R7 interacting with E209^ECL2^ and E297^6.58^ through salt bridges [67].

Structural analysis at the atomic level provides essential knowledge to ascertain the functionality of the receptor. Additionally, in vitro and in vivo pharmacological assessment completes the panorama and offers the necessary insight to develop drugs that offer specific traits with putative application in human therapy. Experiments with recombinant KOP expressed in a human embryonic cell line (HEK-293), and naltrexone competition, receptor internalization, G-protein activation, β-arrestin recruitment, and docking prediction analysis, Ji et al. [77] reported that butorphanol, with a 20-fold higher affinity for KOP, compared with MOP, exhibited a dual role on the KOP activation. It behaved as a partial agonist activating G-protein and a full agonist recruiting β-arrestin and inducing receptor internalization. The butorphanol binding site is very similar to the orthosteric binding site occupied by MP1104 (see Figure 6C). The analysis of the opioid binding sites in MOP and KOP has given clues regarding the importance of the region defined by ECL2 and TM5 for controlling β-arrestin recruitment [78]. The docking manner of the natural compound salvinorin, a diterpenoid furanelactone, to KOP has also generated valuable structural details for designing and evaluating KOP selective and effective drugs [79].

#### 2.6.4. The Nociceptin/Orphanin FQ Receptor (NOP.)

The human NOP is encoded by the gene *OPRL1* (chromosomal location 20q13.33). It has 370 amino acids. Posttranslational processing events of NOP include a disulfide bridge, N-glycosylation, phosphorylation, and lipidation sites (Figure 7A) [49,50].

Nociceptin, a heptadecapeptide, binds and activates NOP, exhibiting a unique functional profile that relates to and distinguishes the receptor system from the considered classical opioid receptors (MOP, DOP, and KOP) [42,81,82]. Unlike the endogenous opioid peptides, a Phe residue, instead of Tyr, appears at its N-terminal end (Figure 2). This feature has consequences regarding binding affinity and receptor selectivity. The architectural traits of NOP and its conformational states support the development of ligands with selected properties [82,83,84].

X-ray diffraction analysis of the complex NOP-SB612111 places the selective NOP antagonist in a cavity where residue D^3.32^ establishes a salt interaction with the ligand [80] (Figure 7B), as it has been observed for other orthosteric opioid-binding pockets DOP and KOP (see Figure 5D and Figure 6D) and for the interaction of MOP D147^3.32^ with the selective inverse agonist alvimopan [85] or with fentanyl [86]. A recent assessment [67] of the structure of the complex NOP-nociceptin shows how the peptide ligand directs its N-terminus toward the bottom of the binding site. Residue F1 connects with Y131^3.33^ through π-π interactions and adapts to a hydrophobic environment shaped by Y131^3.33^, M134^3.36^, I219^5.42^, V279^6.51^, and V283^6.55^ (Figure 7C).

The human ZOP (OGF) protein has 677 amino acids and is encoded by the gene *OGFR* (chromosomal location 20q13.33). Posttranslational processing events of NOP include Ser phosphorylation (Figure 8A) [49,50].

The structure of ZOP is different from the structure of OR. Additionally, the receptor protein localizes in the outer nuclear membrane and internalizes, bound to its natural ligand, [Met5]-ENK (MET) (Figure 8B), to the interior of the cell nucleus through the nuclear pores [22,37,38].

We do not have an accurate three-dimensional atomic-level structure of ZOP. The AlphaFold prediction [87,88] calculates the organization of the protein’s tertiary structure (Figure 8B). However, further analysis is needed to ascertain its role in analgesia, cell growth control, metabolic and immune responses [90,91,92,93], and the mechanism of internalization upon ligand activation and regulation by other proteins, including opioid receptors [70].

### 2.7. The Distribution of OR

The endogenous opioid system, including opioid peptides and receptors, is widely distributed in the central and peripheral nervous system and other tissues (UniProt references P35372 for MOP; P41143 for DOP; P41145 for KOP; P41146 for NOP and Q9NZT2 for ZOP) [19]. Hence, its modulatory role in numerous biological activities, from analgesia to intermediary metabolism or cell growth control. Recent research focuses on OR distribution and capacities related to their function and dysfunction in different pathologies, including cancer (see, for example, [42,94,95,96,97,98,99]).

### 2.8. Signaling Pathways of OR

Figure 9 illustrates representative signaling pathways activated by OR and associated with cancer development and progression (Figure 9).

When opioid ligands bind to the classical OR, conformational changes determine the recruitment of signaling transducers Gi/o heterotrimeric protein or β-arrestin [20,40,74,100,101,102] (Figure 9). Upon G-protein activation, the heterotrimer dissociates. The beta–gamma complex controls ionic potassium and calcium currents through plasma membrane channels, and the alpha subunit bound to GTP inhibits adenylyl cyclase. Activation of β-arrestin may have different outcomes depending on the phosphorylation state of the receptor protein through the kinase activity of GRKs (G-protein coupled receptor kinases) on specific receptor residues [103]. It could lead to receptor internalization, inactivation, recycling, or effective subcellular signaling [104,105,106]. Additionally, receptor internalization of opioid receptors after ubiquitination and independent of Gi/o or phosphorylation mechanisms contributes to the signaling display and deserves exploration in terms of physiological meaning [107].

Activation of ZOP, the opioid growth receptor, by its endogenous ligand, MET, occurs at the cell’s nuclear membrane. The complex receptor-ligand internalizes to the nucleus and interferes with proteins responsible for the operation of the cell cycle [108].

Opioid receptors, adopting multiple structural active states that regulate intracellular action by biased signaling [109] through different transducers, may generate various downstream molecular events achieved with selective ligands. In this line, much basic research supports the pharmacological profile of biased ligands and allosteric modulators for MOP [61,110], DOP [111,112], and KOP [84,113,114]. However, careful revision of biased signaling impact is needed to establish whether some effects observed after ligand-activated receptors have functionally biased significance or may be attributable to the low efficacy of some drugs [115].

As it has been shown for MOP, the up-regulation of specific splice variants may be responsible for the switch of Gs activation instead of Gi/o, amplifying and diversifying the cellular responses dependent on ligand-activated MOP [56]. The diversification of opioid signaling may also source from the transactivation of receptor tyrosine kinases (RTK) that signal through different routes [116,117]. Also, the formation of heteromer structures between OR and OR with other GPCRs may be physiologically relevant and serve as molecular targets of designed drugs [118].

The occurrence of subcellular signaling events after opioid receptor internalization also contributes to the intricate net of OR cellular signaling. This process effectively creates a reduced influence space with distinctive signaling properties [119]. Opioid receptors at the Golgi apparatus activate Gi/o but do not recruit β-arrestin. Also, the receptors signal selectively depending on the lipid composition where OR are immersed [106].

The signaling landscape of OR through direct or interconnected biochemical pathways is complex and malleable. When disrupted, it may provoke alteration of metabolic and gene expression signatures leading to uncontrolled cell growth and migration and causing cancer development or progression [120,121]. Consequently, the analysis of OR signaling in cancer scenarios is paramount to clarify their participation and design new pharmacological strategies that disrupt biochemical cancer pathways.

## 3. Involvement of Opioid Peptides in Cancer

Many studies have demonstrated the involvement of opioid peptides in cancer [10,11,12,13]; these peptides have enhanced the tumor growth induced by stress [122]. MET and dynorphin (DYN) A are released from immune cells under inflammatory conditions [123], and the level of DYN in the cerebrospinal fluid increased in patients with cancer pain [124]. The re-expression of the mu-opioid receptor gene in tumor cells increased the release of beta-endorphin (END) from these cells [125]. Moreover, skin-derived beta-END mediates the fatigue induced by radiation therapy in cancer patients; plasma beta-END level augmented in rats receiving radiation but was reversed with naloxone [126]. The level of plasma beta-END decreased in patients with malignant tumors treated with oxycontin; this treatment also relieved pain and improved clinical symptoms [127]. HSC-3 cells (human tongue squamous cell carcinoma) transfected with the *mu-opioid receptor* gene released more beta-END than control HSC-3 cells, and this virus-mediate gene delivery also attenuated cancer-induced pain [128]. Another study has demonstrated that the administration of flurbiprofen increased the analgesic effects of opioids by increasing plasma beta-END levels in patients showing refractory cancer pain [129].

Moreover, MET, through the opioid growth factor receptor, exerts an antitumoral effect against colon cancer, neuroblastoma, ovarian cancer, head and neck squamous cell carcinoma, and pancreatic cancer [14,15,16,17]. However, other studies have reported that opioids promote tumor growth, metastasis, and vascularization. Because opioids are used to treat pain in cancer patients, the mechanisms involving both dual and opposite actions must be studied in-depth [18].

### 3.1. Bone Cancer

#### Endorphin

Cinobufagin, used for the treatment of cancer pain, promoted beta-END mRNA and protein expressions, but not DYN A, in the microglia of the spinal cord in a rat bone cancer model; this effect was mediated by the alpha7-nicotinic acetylcholine receptor and induced mechanical anti allodynia [130]. Moreover, cinobufagin relieved cancer pain by upregulating the expressions of both mu-opioid receptors and beta-END in the hind paw tumor and tissues placed close to the tumor in an experimental animal model of paw cancer pain [131]. Compared with the sham group, the concentration of beta-END decreased in the rostral ventromedial medulla and spinal cord in an experimental model of cancer-induced bone pain; electroacupuncture and wrist-ankle acupuncture did not affect beta-END concentrations in these regions [132]. Beta-END increased body weight gain, NK (natural killer) cell cytotoxicity, T cell proliferation, and the relative quantities of T cell subtypes but did not affect T cell release in an experimental rat model of bone cancer pain [133].

### 3.2. Brain Tumors

#### 3.2.1. Enkephalin

Pro-enkephalin and MET have been observed in human brain tumors (e.g., ganglioglioma, glioma, meningioma) and associated cyst fluids [134]. Moreover, an inverse correlation between MET and brain tumor malignancy degree has been reported: higher MET level, lower tumor degree [134]. MET promotes apoptosis in rat C6 glioma cells, increases the activity of caspases 3, 8, and 9 and the expressions of Bax, FasL, and Fas, decreases Bcl-2 expression, and increases Ca^++^ influx into the cytoplasm and NFAT1 accumulation into the cell nucleus [135]. No effect on tumor cell viability and FasL upregulation was observed when NFAT1 was knocked down [135]. The results suggest that NFAT1 regulates downstream genes (e.g., *FasL*) and promotes apoptosis in tumor cells. MET-binding sites decrease with increasing malignancy of gliomas, and a shift from mu-opioid receptors in low-grade gliomas to delta-opioid receptors in high-grade gliomas has been reported [136]. These results suggest an inactivation of the MET/opioid receptor system in glioma, which blocks the inhibitory action exerted by MET on average astrocyte growth, promoting tumor progression. Moreover, pro-enkephalin expression was favored by norepinephrine and downregulated by endothelin-1 in C6 rat glioma cells [137,138], and MET favored the growth of human U-373 MG astrocytoma cells [139].

#### 3.2.2. Endorphin

Beta-END-binding sites were reported in the human glioblastoma SF126 cell line [140], and beta-END was observed in human brain tumor cyst fluids [141].

#### 3.2.3. Dynorphin

Glutamate augmented the water content in C6 glioma cells, whereas DYN A_1-13_, via kappa receptors, decreased it [142].

### 3.3. Breast Cancer

A polymorphism in the *mu-opioid receptor* gene has been associated with a reduced response to opioids in breast cancer; patients showing this polymorphism had better survival [143]. MET, LEU, and beta-END expressions have been reported in cells and stroma of human breast cancer samples [144]. The activation of hypoxia-inducible factor 1alpha by delta-opioid receptors promoted cyclooxygenase 2 expression, through phosphatidylinositol 3 kinase (PI3k)/protein kinase B (Akt) stimulation, in breast cancer cells (MCF-7, T47D) causing a paracrine activation of the vascular endothelial cells by prostaglandin E_2_ receptors [18].

#### 3.3.1. Enkephalin

MET, but not LEU stimulated the migration of MDA-MB-468 human breast carcinoma cells [145], and END did not affect the migratory capacity of these cells [145]. A study has reported that the low-fasting plasma level of pro-enkephalin is correlated with an increased risk of breast cancer development in postmenopausal/middle-aged women [146]. The opioid growth factor (MET)/opioid growth factor receptor (receptor zeta) system blocked the proliferation of triple-negative breast cancer cell lines (BT-20, MDA-MD-231), and this effect was mediated by p21 cyclin-dependent inhibitory kinase pathways [147].

#### 3.3.2. Endorphin

High plasma beta-END concentrations have been observed in healthy women, which were even higher in healthy postmenopausal women; however, lower levels were found in women with breast cancer, and no difference was observed between premenopausal and postmenopausal women who have breast cancer [148]. Moreover, chemotherapy only improved beta-END levels in postmenopausal women but without reaching the levels observed in healthy women [148]. Beta-END activates the survival/mitogenic signaling pathways (Akt, signal transducer and activator of transcription 3 (STAT3) and mitogen-activated protein kinases (MAPK)/extracellular signal-regulated kinase (ERK)) in human MDA-MB-231 breast cancer cells. Increasing plasma beta-END levels have been correlated with increasing tumor burden in a mouse model of breast cancer [149]. This observation means that the peptide is involved in cancer progression, and, significantly, the high levels of plasma beta-END did not decrease pain in mice with breast tumors; quite the opposite, the pain increased in these animals [149]. Patients with breast carcinoma treated with a galactose-specific lectin standardized mistletoe extract showed an increase in the level of plasma beta-END and an enhancement of T lymphocytes and NK cells [150]. A correlation between plasma beta-END level and the activity of the last two cells was also reported [150] and between an increased plasma beta-END level and an improved quality of life in patients with breast cancer [151].

In female rats in which a mammary tumor was developed, the level of beta-END was higher in the midbrain, striatum, and pituitary; the level of MET was lower in the striatum, and that of DYN decreased in the hypothalamus and pituitary when compared with the levels observed in control animals [152]. Fetal alcohol exposure reduces beta-END levels, causing a hyper-stress response and inhibiting the immune action against tumors [153]. Fetal alcohol exposure and control rats treated with N-nitroso-N-methylurea to promote mammary cancer growth were studied, and neurons expressing beta-END were transplanted into the hypothalamus after tumor development to augment beta-END production [153]. This strategy blocked tumor development in fetal alcohol-exposed and control animals and reversed the effects mediated by fetal alcohol exposure regarding the susceptibility to breast cancer. In this sense, it has been suggested that beta-END regulates the stress response and promotes innate immunity preventing breast cancer development [154].

Moreover, it seems that the cancer-preventive effect mediated by beta-END was due to an inhibition of the sympathetic neuronal action, which increased the activities of NK cells and macrophages and the production of anti-inflammatory cytokines [155]. Thus, beta-END counteracts breast cancer development by favoring immune-mediated anticancer defenses [156,157]. In addition, beta-END alters the tumor microenvironment by inhibiting the production of catecholamines and inflammatory cytokines, which alter cell-matrix attachment, DNA repair, epithelial-mesenchymal transition, and angiogenesis [155].

#### 3.3.3. Dynorphin

The presence of DYN A_1-17_ and DYN A_1-8_ has been reported in Walker 256 tumors, a carcinosarcoma originated from the mammary gland of rats, but no opioid binding site was observed [158].

### 3.4. Cervical Cancer

#### 3.4.1. Enkephalin

MET blocked cervical carcinoma progression in vivo; decreased myeloid-derived suppressor cell-infiltrated both tumor and circulation; induced apoptosis, and increased the expressions of caspase 3 and 8, Fas, and the signaling pathway mediator Bax [159]. This result suggests that MET is a promising antitumor research line in cervical cancer.

#### 3.4.2. Endorphin

Electroacupuncture increased plasma beta-END levels in cervical cancer patients [160].

### 3.5. Colorectal Cancer

A low-dose of naltrexone (an opioid antagonist) blocked colorectal cancer progression in vivo and in vitro [161]. This treatment augmented the expressions of macrophage markers (CD68, F4/80), M1 macrophage phenotypic markers (CD80), and the level of cytokines (tumor necrosis factor-alpha) [161]. Moreover, a low-dose of naltrexone upregulated the expressions of the opioid growth factor receptor and apoptotic factors (PARP, caspases 3 and 9, Bax) and downregulated the expressions of Ki67 and Bcl-2, causing apoptosis in tumor cells [161].

#### 3.5.1. Enkephalin

The presence of enkephalin has been reported in rectal carcinoids [162]. MET enhanced in vivo colon carcinogenesis induced by azoxymethane [163]. However, MET exerted an antitumor action in vivo against colorectal tumors (MC38 cell line) by acting on the tumor microenvironment and the immune system [164]. MET augmented both immunogenicity and recognition of tumor cells; downregulated Kras (oncogene), Bcl2, and Bclxl (anti-apoptotic proteins); blocked the synthesis of inflammatory cytokines; reduced immune checkpoints (2b4, Flgl1, Lag3, Pd-11, Pd-1) in tumor cells, and increased CD4^+^T, CD8^+^T, and macrophages (M1) infiltration [164]. The antitumor effect exerted by MET was also mediated by effector T cells; the peptide upregulated the opioid growth factor receptor, and the specific inhibitor of this receptor, naltrexone, blocked the antitumor action induced by MET [164]. The data support that MET is a promising therapeutic agent against colorectal cancer by improving immunotherapy efficacy. Moreover, CD10 (a marker for liver metastasis in colorectal cancer) increased colorectal cancer cell metastasis by abrogating the antitumor action mediated by MET since the peptide blocked the growth, invasion, and survival of tumor cells after thiorphan (inhibitor of the enzyme that degrades enkephalins)-induced CD10 inactivation [165]. MET, via delta opioid receptors, decreased the phosphorylation of ERK/epidermal growth factor receptors and increased p38-dependent apoptosis [165]. LEU decreased the invasive capacity of murine colon 26-L5 adenocarcinoma cells [166].

#### 3.5.2. Endorphin

Beta-END has been reported in adenocarcinomas derived from the colon mucosa; its expression was higher in adenocarcinomas than in the mucosal layer of normal colons [167], and the peptide has also been observed in rectal carcinoids [162]. The effects of ultraviolet A eye irradiation on colon carcinoma induced by dextran sodium sulfate and azoxymethane have been studied in an experimental mouse model [168]. Mu-opioid receptors, MET, and beta-END expressions increased in treated animals, and these expressions increased more when these animals received ultraviolet A eye irradiation [168]. Colon carcinoma symptoms were decreased after this irradiation, but these beneficial effects were reduced when beta-END inhibitors were administered and disappeared with naltrexone [168]. This fact suggests that the beneficial effects observed in colon carcinoma after the ultraviolet A eye irradiation were mediated by beta-END and MET. Hypothalamic neurons containing beta-END inhibited the development of preneoplastic/neoplastic lesions in an experimental colon cancer model induced by 1, 2-dimethylhydrazine [169]. Animals with hypothalamic beta-END neuronal transplants (70%) failed to develop tumors, and animals with transplanted beta-END neurons showed a lesser adenoma development in the colon and tissue lesions (e.g., aberrant crypt foci) and decreased expressions of Ki-67, tumor necrosis factor-alpha, and NF-κB nuclear translocation in colonic tissues [169].

Moreover, decreased levels of transcription factors linked to epithelial-mesenchymal transition (e.g., Twist, Snail, N-cadherin) and increased levels of E-cadherin were observed in the colon tissue of transplanted animals [169]. The data suggest that beta-END neuron transplants inhibited colon cancer progression by reducing the epithelial-mesenchymal transition and the inflammatory mechanisms. However, the administration of beta-END into the nucleus of the raphe magnus induced analgesia. It facilitated the metastasis, which was inhibited with naloxone, and when this nucleus was electrically stimulated and analgesia induced, the metastasis was considerably attenuated [170].

#### 3.5.3. Dynorphin

MET, DYN A_1-8_, beta-END, and LEU (10^−4^ M; 10^−6^ M) did not affect the migration, chemotaxis, or invasion of colorectal tumor cells (HCT-116, HT-29) [171]. Moreover, MET, DYN A_1-8,_ and beta-END (10^−6^ M) did not alter the viability of the HT-29 tumor cell line [172].

### 3.6. Cutaneous Squamous Cell Carcinoma

#### Enkephalin

MET blocked the cell growth of cutaneous squamous cell carcinomas by inducing the G0/G1 cell cycle arrest and by promoting apoptotic mechanisms through the caspase 3/Bax/Bcl-2 signaling pathway [173]. MET, through the opioid growth factor receptor, blocked the proliferation of A431 cells by inducing apoptosis, promoting autophagy in cutaneous squamous cell carcinoma cells, and activating dendritic cells [174]. MET also decreased immunosuppression by reducing the number of myeloid-derived suppressor cells, controlling the polarization of tumor-associated macrophages from M2 to M1, and inhibiting the JAK2/STAT3 tumor-promotion/immunosuppression signaling pathway, which is involved in macrophage polarization and myeloid-derived suppressor cell expansion [173]. Thus, MET exerts an antitumor effect against cutaneous squamous cell carcinoma.

### 3.7. Gastric Cancer

#### 3.7.1. Enkephalin

MET blocked the growth of human gastric cancer HGC27 and SGC7901 cell lines [175]. The peptide arrested the cell cycle in the G0/G1 phase, reducing Ki67, cyclin D1, and c-myc mRNA and promoting apoptosis by upregulating the Bax expression through downregulating Bcl-2/surviving expressions and by activating PARP and caspase 3 [175]. MET also upregulated the expression of the opioid growth factor receptor. Another study has demonstrated that MET-induced apoptosis in human gastric tumor cells (HGC27, SGC7901) by inhibiting the PI3k/Akt/mammalian target of rapamycin (mTOR) signal pathway, reduced the number of M2-type macrophages and increased the M1-type [176]. Tumor cell apoptosis was blocked when the opioid receptor expression was knocked down [176]. The data suggest that MET is a promising antitumor agent against gastric cancer.

#### 3.7.2. Endorphin

Plasma beta-END levels decreased in gastric cancer patients after transcutaneous electrical acupoint stimulation [177], and beta-END was observed in adenocarcinomas derived from the antral mucosa [178].

### 3.8. Head and Neck Cancer

The activation of the mu-opioid receptor promoted head and neck squamous cell carcinoma growth in vitro and in vivo experiments, increasing the proliferation and migration of tumor cells (FaDu, MDA6868Tu) [179]. Thus, the mu-opioid receptor is a promising antitumor target to treat head and neck squamous cell carcinomas.

#### 3.8.1. Enkephalin

MET expression has been reported in human head and neck squamous cell carcinomas [180]. Reduced DNA synthesis and tumor weight/volume have been reported in the head and neck squamous cell carcinomas after treatment with MET, imiquimod, and naltrexone (low dose) [181]. The inhibitory action exerted by the MET/opioid growth factor receptor system is mediated through the p16 pathway; MET increases the expression of the cyclin-dependent kinase inhibitor p16 protein [182]. Downregulation of the opioid growth factor receptor favored the progression of head and neck squamous cell carcinoma [183], and LEU has been detected in head and neck paragangliomas [184].

#### 3.8.2. Endorphin

Beta-END increased the production of the leukocyte migration inhibitory factor, reaching almost normal levels, in patients with squamous carcinoma of the head and neck; this means that the peptide regulates the immune system [185].

#### 3.8.3. Dynorphin

MET, DYN A_1-8_, beta-END, and LEU (10^−4^ M; 10^−6^ M) did not affect the migration, chemotaxis, or invasion of squamous carcinoma cells (CAL-27, SCC-1) [186]. Moreover, MET, DYN A_1-8,_ and beta-END (10^−6^ M) did not affect either the differentiation of SCC-1 and CAL-27 tumor cells [187] or the viability of the latter cells [172].

### 3.9. Larynx Cancer

#### 3.9.1. Enkephalin

MET has been observed in neuroendocrine tumors (paragangliomas) of the larynx [188].

#### 3.9.2. Endorphin

Beta-END has been reported in tumor cells in an oat cell carcinoma of the larynx [189].

### 3.10. Leukemia

#### 3.10.1. Enkephalin

The presence of pro-enkephalin A has been reported in leukocytes from patients with chronic lymphoblastic leukemia [190], and MET increased CD10 expression and inhibited the metabolic activity of the leukemic NALM-1 cell line [191]. MET promoted apoptosis in K562 human erythroid leukemia cells [192], and the peptide favored pre-B acute lymphoblastoid cell migration (LAZ 221, NALM 6) and increased the CD9 surface expression (a leukemia cell marker) in the latter cells [193]. This migration, induced by MET, was considerably decreased when pre-B acute lymphoblastoid cells were incubated with antibodies against CD9 [193].

#### 3.10.2. Endorphin

Beta-END has been reported in the cerebrospinal fluid of children with acute lymphoblastic leukemia; the highest level was observed at the end of the intensification chemotherapy, whereas treatment with glucocorticoids decreased beta-END levels [194]. However, plasma beta-END levels decreased in patients with solid tumors after treatment with the chemotherapeutic drug CDDP [195]. Plasma beta-END levels were higher in patients with acute leukemia than in healthy individuals, and stress factors (e.g., high temperature, anemia, hypoxic conditions, pain, cancer) increased the synthesis of beta-END, particularly in the white blood cells of patients with acute leukemia during chemotherapy treatment [196]. Beta-END promoted the growth of T-lymphoblastoid cells; however, this was not observed in promyelocyte and B-lymphoblastoid cells [197]. Finally, naloxone-resistant receptors for beta-END are downregulated after activation of the protein kinase C in the U937 cell line (isolated from a histiocytic lymphoma) [198].

### 3.11. Liver Cancer

#### 3.11.1. Enkephalin

The proliferation of hepatocellular carcinoma cells (Hep 3B, Hep G2, SK-HEP-1) was inhibited after treatment with MET due to the blockade of the DNA synthesis and not to necrotic/apoptotic mechanisms [199]. Moreover, silencing the opioid growth factor receptor promoted the proliferation of these cells [199]. MET concentration was higher in metastasis-positive human livers than in normal ones [165]. This finding is important since it suggests that the increase in MET could be an endogenous antitumor mechanism to counteract tumor progression. Two patients with hepatoblastoma were cured after surgical resection and treatment with naltrexone (low dose) and MET [200]. It seems that this treatment is a promising antitumor therapeutic strategy.

#### 3.11.2. Endorphin

In an experimental animal model of liver cancer, neurons expressing beta-END transplanted into the hypothalamus prevented hepatocellular carcinoma formation and hepatocellular injuries [201]. This strategy inhibited carcinogen-induced liver histopathologies (e.g., fibrosis, collagen deposition, inflammatory infiltration) and augmented the concentration of NK cell cytotoxic agents in the liver [201].

### 3.12. Lung Cancer

#### 3.12.1. Enkephalin

Pro-enkephalin and MET have been reported in human lung cancer cells [202], and serum LEU level was higher in patients with bronchial carcinoma than in control individuals [203]. Morphine, via the opioid growth factor receptor, suppressed lung cancer cell proliferation (H1975); this suppression occurred in the cell cycle S phase [204]. By controlling the Wnt/beta-catenin pathway, MET exerted an antitumor effect against lung cancer in vitro and in vivo experiments, leading to cell cycle arrest at the G0/G1 phase [205]. Moreover, the antitumor action of the peptide was abolished in the knockdown of growth factor receptor, and MET augmented the infiltration of dendritic cells, CD4^+^ T and CD8^+^ T cells, macrophages (M1) and NK cells, and reduced the number of macrophages (M2) and myeloid inhibitory cells [205]. MET also upregulated the expression of interleukin-15, interleukin-21, interferon-gamma and downregulated interleukin-10, and tumor necrosis-beta 1 expression in the tumor microenvironment [205]. MET increased the expression of the opioid growth factor receptor and, by activating the caspase 3/Bax/Bcl-1 signaling pathway, promoted apoptosis in lung cancer cells [206]. These effects disappeared when the previous receptor was blocked. MET also increased the immunogenicity of lung cancer cells, NK cell activity, and the expression of NK cell-related cytokines (e.g., interferon-gamma, granzyme B) [206]. Previous findings suggest that MET is a promising antitumor agent against lung cancer. However, another study has demonstrated that methylnaltrexone (an opioid antagonist) counteracted Lewis lung carcinoma growth (cancer cells express mu-receptors) and decreased lung metastasis and that morphine (a mu-receptor agonist) favored tumor growth [207]. It is important to note that nicotine partially or reversed opioid-induced growth suppression in 9/14 lung cancer cell lines studied [208]. Moreover, MET blocked pulmonary metastasis and enhanced the activity of NK cells [209].

#### 3.12.2. Endorphin

Beta-END has been reported in the bronchoalveolar lavage fluid of patients with lung cancer [210] and in the plasma of patients with this disease [211]. Beta-END has been reported in lung small-cell carcinomas and carcinoid tumors [212]. Human small-cell lung carcinoma cell lines (NCI-N417, NCI-H345, NCI-H69) express naloxone-insensitive endorphin binding sites that were insensitive to naloxone and other mu-, delta- or kappa-opioid receptor ligands [213]. The U1,690 cell line (small-cell lung carcinoma) expresses beta-END, and the peptide promotes the proliferation of this cell line through non-opioid binding sites; moreover, beta-END binding did not affect the synthesis of cAMP [214]. Beta-END also acts as a chemoattractant for small-cell lung carcinoma cells favoring migration and metastasis [215].

#### 3.12.3. Dynorphin

Pre-proDYN mRNA has been reported in small-cell lung carcinomas [216], and serum DYN A/B and MET levels increased in a non-small-cell lung cancer cell xenograft stress reduction mouse model [217]. Moreover, via Gαi, DYN B blocked cAMP formation in non-small-cell lung cancer cells [217]. Most lung tumor cells co-expressed pro-DYN and DYN, prohormone convertase 1, prohormone convertase 2, or carboxypeptidase E. In contrast, a few lung cancer cells only expressed one of the last markers [218]. DYN was observed in cancer cells infiltrating human lung tissues and nerve fibers in the bronchial submucosa [218]. Lung cancer cells express mu, kappa, and delta opioid receptors and contain several combinations of opioid peptides (DYN, ENK, beta-END). After opioid administration, cAMP concentration was decreased in these cells [208]. In addition, agonists of the previous three receptors (1-100 nM) blocked the growth of tumor cells in vitro, whereas nicotine (100-200 nM) totally or partially counteracted the growth blockade mediated by opioids [208].

*Pro-opiomelanocortin gene* delivery blocked the growth of alpha-melanocyte-stimulating hormone/melanocortin 1 receptor (MC1R)-deficient Lewis lung carcinoma cells as well as the growth of these cells in mice; apoptotic mechanisms mediated these effects through an MC1R-independent pathway [219]. The authors also demonstrated that, in this case, via an MC1R-dependent pathway, this delivery blocked tumor progression and metastasis of B16-F10 melanoma cells [219]. *Pro-opiomelanocortin* gene delivery attenuated the beta-catenin signaling pathway by decreasing protein concentrations of beta-catenin and its downstream proto-oncogenes (e.g., *c-myc, cyclin D1*) and blocked tumor vasculature [219].

### 3.13. Melanoma

#### 3.13.1. Enkephalin

Compared to normal skin, the expression of MET and LEU was decreased in melanocytic tumors [220]. MET and LEU have been detected in six of seven secondary neuroendocrine carcinomas of the skin, whereas both peptides were not found in skin primary neuroendocrine carcinomas (Merkel cell carcinoma) [221]. MET exerted an antitumor effect in mice xenografted with melanoma B16-BL6 cells which was inhibited with naloxone; the antitumor action was due to the immune system’s modulation and a cytotoxic effect on melanoma cells [222]. In the same experimental model, MET promoted cell cycle arrest. It increased the plasma levels of interferon-γ, tumor necrosis factor-alpha, and interleukin-2 [223]. MET promotes cell cycle arrest in the G0/G1 phase, decreases the number of cells in the S and G2/M phases, and increases the expression of the opioid growth factor receptor in B16 melanoma cells [223]. Tumor growth and tumor cell dissemination were counteracted with MET, and the peptide blocked A375 melanoma cell proliferation through apoptotic mechanisms [223,224]. MET also promoted cell cycle arrest in the G0/G1 phase, decreased the cell number in S and G2/M phases, and favored apoptosis in human A375 melanoma cells [224]. Imiquimod also upregulates the expression of the opioid growth factor receptor, increasing MET’s antitumor effect [225]. This synthetic immune response modifier has been successfully applied in melanoma treatment [226,227,228,229,230]. MET did not affect adenylate cyclase activity in AB16 melanoma cells [231]. Moreover, MET decreased tumor weight and volume in vivo and increased the ratio of CD4^+^ to CD8^+^ T cells [223].

#### 3.13.2. Endorphin

Beta-END was observed in 30 of 42 melanoma samples [232]. Beta-END was found in six of seven secondary neuroendocrine carcinomas of the skin but was absent in primary neuroendocrine carcinomas (Merkel cell carcinoma) [221]. B16 melanoma cells synthesize and release beta-END [233]. Tumor growth was studied in mu-opioid receptor-deficient and wild-type mice administered with B16 melanoma cells producing beta-END [233]. B16 cells decreased tumor growth and increased the infiltration of immune cells into the tumor in mu-opioid receptor-deficient animals. Opioids in the B16 cell supernatant reduced the proliferation of normal leukocytes but not those from mu-opioid receptor-deficient animals [233]. A correlation was observed between beta-END levels and tumor progression in melanoma tissues [233]. In this sense, beta-END immunoreactivity was lower in benign melanocytic naevi than in metastatic and advanced melanomas [232]. Moreover, a low-dose ultraviolet exposure promoted beta-END synthesis in epidermal keratinocytes and increased the plasma level of the peptide [234]. Thus, mu-opioid peptides modulate the immune response and control the development of tumors. Moreover, beta-END did not affect adenylate cyclase activity in AB16 melanoma cells [231].

### 3.14. Myeloma

#### Dynorphin

U50,488, a kappa-opioid receptor agonist, favored Fas-induced apoptosis without Fas receptor expression increase and decreased cell proliferation in human multiple myeloma LP-1 cells expressing mu- and kappa-opioid receptors [235]. However, this study demonstrated that the antiproliferative effect mediated by U50,488 was independent of the kappa receptor. This effect was due to a G0/G1 phase blockade, and cell cycle inhibitors (e.g., p53, p27Kip1, p21Cip1) were not upregulated [235]. DYN or morphine did not regulate apoptosis or cell proliferation in LP-1 cells [235].

### 3.15. Neuroblastoma

#### 3.15.1. Enkephalin

MET has been located in mouse neuroblastoma Neuro2a cells, and the peptide was released from these cells with a high K^+^ stimulation [236]. MET arrested the growth of human SK-N-SH neuroblastoma cells [237].

#### 3.15.2. Endorphin

Neuroblastoma Kelly, NMB, and IMR-32 cell lines express Beta-END-binding sites [238]. Tumors of transplanted neuroblastoma S20Y cells in mice and treated with complete or intermittent opioid receptor blockade with naltrexone showed an upregulation of beta-END and MET levels and MET-binding sites [239]. In this study, MET decreased the tumor mitotic index, which was counteracted with naltrexone. Thus, a complete blockade with naltrexone increased tumor cell proliferation, whereas an intermittent blockade inhibited cancer cell proliferation [239].

#### 3.15.3. Dynorphin

Pre-pro-DYN mRNA and pre-pro-ENK have been reported in neuroblastoma SK-N-MC and SCLC H69 cell lines [216] and opioid binding sites in the neuroblastoma S20Y cell line [240]. A cysteine protease-degrading DYN A_1-13_ and DYN A_1-17_ has been reported in the membrane of mouse neuroblastoma N18 cells [241]. DYN exerted a cytotoxic action, promoted apoptosis, and downregulated the expression of the anti-apoptotic protein Bcl-2 in SH-SY5Y neuroblastoma cells; previous effects were inhibited with the anesthetic isoflurane [242]. MET, DYN A_1-8_, and beta-END did not affect the differentiation of SK-N-SH neuroblastoma cells at the concentration administered (10^−6^ M) [187]. DYN A, at high concentration, binds to neuropeptide Y receptors (Y1 and Y2) in SK-N-MC and SMS-MSN neuroblastoma cell lines, and it has been suggested that DYN A could exert an antagonistic effect on these cells [243]. Moreover, this binding was not mediated either by changes in receptor-G protein interaction or by receptor phosphorylation.

### 3.16. Ovarian Cancer

#### 3.16.1. Enkephalin

Enkephalin has been reported in tumor cells in ovarian carcinoids [244] and MET and the opioid growth factor receptor in human ovarian cancer cells [245]. MET, in a dose-dependent manner, exerted an inhibitory proliferative effect on ovarian tumor cells (HEY, CAOV-3, SW626), whereas the neutralization of MET promoted cell proliferation and the silencing of the opioid growth factor receptor favored tumor cell replication; MET, via the opioid growth factor receptor, delayed cells moving by upregulating the cyclin-dependent inhibitory kinase pathways [16,245]. A low dose of naltrexone inhibited ovarian tumor progression and, in combination with cisplatin, exerted an enhanced inhibitory effect [246].

#### 3.16.2. Endorphin

Beta-END has been observed in ovarian sex cord-stromal tumors [247] and ovarian carcinoids [244]. A positive correlation has been reported between survival time/disease-free time and plasma beta-END level in patients with ovarian cancer [248]. Lower beta-END concentrations were observed in patients with recurrence than those without recurrence [248]. Moreover, beta-END and MET blocked the proliferation of human ovarian KF cancer cells; this effect was counteracted with naloxone, and it seems that both peptides blocked protein/RNA synthesis but not DNA synthesis [249].

### 3.17. Pancreatic Cancer

#### 3.17.1. Enkephalin

A lipid conjugation of MET increased the tumor-suppression activity of the peptide against human pancreatic adenocarcinomas [250], and the MET/opioid growth factor receptor system increased the cyclin-dependent kinase inhibitor p21 protein expression to attenuate the progression of human pancreatic cancer [251]. Moreover, the administration of MET ameliorated clinical symptoms and survival in patients with advanced pancreatic cancer [252], and a high plasma level of MET has been observed in patients with pancreatic cancer [253].

#### 3.17.2. Dynorphin

DYN A1-8, MET, beta-END, and LEU (10^−4^ M; 10^−6^ M) did not affect the migration, chemotaxis, or invasion of pancreatic tumor cells (MIA PaCa-2, PANC-1, BxPC-3) [186] and the first three peptides mentioned did not alter the viability of the MIA PaCa-2 tumor cell line at the concentration administered (10^−6^ M) [172].

Mouse insulinoma beta TC3 cells show a high expression of pro-DYN mRNA and its derived peptides (DYN A_1-8_, DYN B_1-13_, alpha-neo-END) [254]. These cells also expressed prohormone convertase 1 and 2 mRNAs but not convertase 5 mRNA, and cells administered with 8-bromo-cAMP increased pro-DYN levels and the release of opioid peptides [254].

### 3.18. Pheochromocytoma

#### 3.18.1. Enkephalin

Proprotein convertase 2 and pro-enkephalin have been reported in human pheochromocytomas [255]. MET has been observed in human pheochromocytomas, and nicotine promoted the release of the peptide from cultured pheochromocytoma cells [256,257]. LEU was also observed in pheochromocytomas [257]. MET and LEU levels differed in extramedullary and medullary tumors: the MET to LEU ratio was higher in extramedullary than in medullary pheochromocytomas [258].

#### 3.18.2. Endorphin

Beta-END has been reported in pheochromocytomas [257,259], and the release of the peptide has also been demonstrated [260].

#### 3.18.3. Dynorphin

The rat pheochromocytoma PC12 cell line expresses the *pro-DYN* gene and releases DYN [171]. The presence of DYN, but not alpha-neo-END, has been reported in human pheochromocytomas [257,261]; however, another study has shown the presence of both DYN and alpha-neo-END in these tumors and, in addition, it was demonstrated that nicotine favored the release of both peptides from pheochromocytomas [262]. In another study, MET, LEU, beta-END, and DYN were reported in all the pheochromocytomas studied in which the concentration of enkephalins was higher than that of DYN, and the DYN level was higher than that reported for beta-END [257]. DYN A_1-17_ was the major component observed in pheochromocytomas, whereas DYN A_1-13_ and DYN A_1-12_ were minor components in these tumors [263,264].

### 3.19. Pituitary Cancer

#### 3.19.1. Enkephalin

MET level was increased in prolactin-releasing human pituitary adenomas [265].

#### 3.19.2. Endorphin

Beta-END was observed in a pituitary adenoma [266], and the peptide was released from human pituitary cancer cells [267]. The presence of beta-END has been reported in clinically silent pituitary corticotroph adenomas [268], and beta-END and beta-END_1-27_ have been found in extracts of pituitary melanotroph tumors transplanted subcutaneously in mice [269]. W7, a calmodulin inhibitor, potentiated beta-END release promoted by 8-BcrAMP from the mouse anterior pituitary AtT-20 cancer cell line [270]. Cerebrospinal fluid beta-END levels increased after resectioning an adrenocorticotropic hormone-secreting pituitary adenoma; however, the MET level was not altered [271].

### 3.20. Prostate Cancer

The expression of opioid receptors has been described in prostate cancer cells and tissues [272]. Zeta-opioid receptor mRNA was expressed in all the prostate cancer cell lines studied, kappa-opioid receptors in only two cell lines (VCaP, LNCaP), and no expression was observed for mu- and delta-opioid mRNA receptors [272]. Compared with other prostate cancer cell lines, LNCaP (an androgen-sensitive cell) showed a higher expression of kappa- and zeta-opioid receptors and a synthetic androgen (R1881) repressed mRNA of both receptors [272]. Moreover, zeta-opioid receptors showed a higher expression in prostate cancer tissues than in normal ones, and this expression was elevated in aggressive and undifferentiated prostate cancer tissues [272]. A high mu-opioid receptor expression has been associated with poorer survival in patients with prostate cancer [273].

#### 3.20.1. Enkephalin

DAGO ([D-Ala^2^, N-Me-Phe^4^-Gly-ol] enkephalin), DADLE ([D-Ala2, D-Leu5] enkephalin), and DSLET ([D-Ser^2^, Leu^5^] enkephalin) blocked the proliferation on human prostate cancer cell lines (PC3, DU145, LNCaP); this effect was counteracted with the opioid antagonist diprenorphine [274].

#### 3.20.2. Endorphin

Beta-END and LEU have been reported in prostatic carcinomas [275]. Rats with transplanted neurons expressing beta-END into the hypothalamic paraventricular nucleus showed a protective effect against prostate cancer development; an increased NK cell cytolytic action in the spleen and peripheral blood mononuclear cells; a decreased level of inflammatory cytokines (tumor necrosis factor-alpha) in plasma, and a higher level of anti-inflammatory cytokines (interferon-gamma) in plasma [276]. This observation indicates that the release of beta-END from the hypothalamic transplanted neurons counteracts the inflammatory mechanisms and increases the immune system’s response.

#### 3.20.3. Dynorphin

DYN A, DYN A_1-13_, and DYN A_1-7_ promote the growth of the DU145 prostatic carcinoma cell line [277]. Naloxone blocked the effect mediated by DYN A but increased the growth of tumor cells at a high concentration (10^−7^ M). Electroacupuncture counteracted bone-cancer-promoted hyperalgesia in a rat model that received AT-3.1 prostate cancer cells into the tibia [278]. Electroacupuncture blocked DYN and pre-pro-DYN mRNA upregulation, and the administration of anti-DYN A_1-17_ antisera also blocked hyperalgesia [278]. In this sense, an upregulation of DYN A has been reported in the spinal dorsal horn, and the release of the peptide has been related to spontaneous pain in a mouse model of neuropathic cancer pain [279]. Moreover, in a murine model of cancer-pain, the number of immunoreactive neurons containing DYN was increased in the spinal cord (ipsilateral to the limb containing the tumor) [280].

### 3.21. Renal Cancer

#### Enkephalin

MET blocked the proliferation of human renal cancer cells (Caki-2) [281].

### 3.22. Retinoblastoma

#### 3.22.1. Enkephalin

MET has been reported in human retinoblastoma [282].

#### 3.22.2. Endorphin

Beta-END-binding sites have been reported in the human retinoblastoma McA and Y79 cell lines [283].

### 3.23. Testicular Cancer

#### Dynorphin

Pro-DYN mRNA and its derived peptides have been observed in the R2C rat Leydig tumor cell line [284].

### 3.24. Thymic Cancer

#### 3.24.1. Enkephalin

MET has been reported in a thymic carcinoid [285].

#### 3.24.2. Endorphin

Beta-END binds to non-opioid binding sites expressed in thymoma cells [286], and beta-END has been observed in an oncocytic carcinoid tumor of the thymus [287].

#### 3.24.3. Dynorphin

A released dibasic cleaving peptidase that converts DYNs (e.g., DYN A_1-12_, DYN A_1-9_, proDYN B) into LEU-Arg^6^ has been obtained from the medium of EL-4 mouse thymoma cells [288].

### 3.25. Thyroid Cancer

#### 3.25.1. Enkephalin

Human anaplastic thyroid cancer cells (KAT-18) express MET and the opioid growth factor receptor; MET blocked cell replication, the opioid antagonist naltrexone promoted cell growth, and anti-MET antibodies counteracted the inhibitory action mediated by MET [289].

#### 3.25.2. Endorphin

Opioid peptides (e.g., beta-END, alpha-neo-END) derived from the three opioid precursors have been reported in human thyroid medullary carcinomas [11], and the release of beta-END has been demonstrated from cultured medullary thyroid carcinoma cells [290].

Table 1, Table 2 and Table 3 and Figure 10, Figure 11 and Figure 12 show the involvement of MET, LEU, beta-END, and DYN in cancer.

## 4. Antitumor Therapeutic Strategies

The use of opioid peptides as antitumor drugs must be developed; much data support this potential therapeutic strategy. MET blocks the proliferation of triple-negative breast cancer cells through the p21 cyclin-dependent inhibitory kinase pathway [147], and the peptide induces apoptosis in tumor cells and inhibits cervical carcinoma progression [159]. MET also exerts an antitumor action in vivo against colorectal tumors by acting on the tumor microenvironment and improving immunotherapy efficacy [164]; MET inhibits the cell growth of cutaneous squamous cell carcinoma by promoting apoptotic mechanisms and decreasing immunosuppression [173], and MET blocks the growth of human gastric cancer cell lines by promoting apoptosis [175]. Moreover, morphine and MET exert an antitumor effect against lung cancer cells [204,205], and the antitumor action mediated by MET was abolished in the knockdown of growth factor receptors [205]. MET promotes apoptosis in lung cancer cells and increases the expression of the opioid growth factor receptor, but these effects disappear when the receptor is blocked [206]. MET inhibits pulmonary metastasis and increases NK cell activity [209], exerts an antiproliferative action against ovarian tumor cells (HEY, CAOV-3, SW626), delays cell moving [16,245], and blocks the proliferation of human renal cancer cells (Caki-2) [281]. Human anaplastic thyroid cancer cells (KAT-18) express MET; the peptide blocked cell replication, naltrexone promoted cell growth, and anti-MET antibodies counteracted the inhibitory action mediated by MET [289]. MET neutralization favored cell proliferation, and silencing the opioid growth factor receptor promoted tumor cell replication. Beta-END and MET blocked the expansion of human ovarian KF cancer cells; this was counteracted with naloxone, and it seems that both peptides blocked protein/RNA synthesis but not DNA synthesis [249]. MET exerts an antitumor effect, through apoptotic mechanisms, against melanoma cells which were inhibited with naloxone [222,223,224]. MET and imiquimod upregulate the expression of the opioid growth factor receptor, increasing MET’s antitumor effect [223,225].

Moreover, DAGO, DADLE, and DSLET inhibited the proliferation of human prostate cancer cell lines (PC3, DU145, LNCaP), which was counteracted with the opioid antagonist diprenorphine [274]. Thus, MET exerts an antitumor action against cervical carcinoma, colorectal tumor, cutaneous squamous cell carcinoma, and breast, gastric, lung, ovarian, renal, and thyroid cancers. The observation implies that MET is a promising broad-spectrum antitumor agent; this potential and good antitumor strategy must be developed.

By contrast, opioids promoted the growth of MCF-7 breast cancer cells, which was blocked with the mu-opioid receptor antagonist naloxone, and naltrexone promoted apoptosis in a mouse model of MCF-7 cells and decreased the growth of the triple-negative breast cancer MDA-MB-231 cell line [147,291,292]. Naltrexone also inhibited ovarian tumor progression and, with cisplatin, exerted an enhanced inhibitory effect [246]. A low dose of naltrexone blocked colorectal cancer progression in vivo and in vitro by inducing the apoptosis of tumor cells [161]. Methylnaltrexone (mu-opioid receptor antagonist) increased the activity of the antitumor drug docetaxel by inhibiting gastric cancer cell growth-suppressive pathways in vivo [293]. Thus, it seems that the blockade of the pathways that inhibit cell growth increases the actions of the antitumor agents. Methylnaltrexone had antimetastatic and antiproliferative effects against lung carcinoma [207]. Methylnaltrexone also inhibited the angiogenesis mediated by opioids. It blocked the proliferation/migration of the endothelial cells mediated by opioids by inhibiting the vascular endothelial growth factor (VEGF) receptor phosphorylation/transactivation, leading to the blockade of RhoA activation [294]. Methylnaltrexone counteracted Lewis lung carcinoma growth and decreased lung metastasis, and the mu-receptor agonist morphine favored tumor growth [207]. MET declined the tumor mitotic index, which was counteracted with naltrexone: a complete blockade with naltrexone increased neuroblastoma cell proliferation, whereas an intermittent blockade inhibited cancer cell proliferation [239]. Beta-END favors small-cell lung carcinoma cells’ proliferation, migration, and metastasis [215]. Opioid peptides (e.g., MET) facilitate the migration of tumor cells [145]; this means that opioid peptide-receptor antagonists could exert an antimetastatic action. This antitumor strategy must be fully exploited.

Thus, opioids exert a dual action on tumor cells: a proliferative or antiproliferative activity. This observation illustrates that opioid peptides (e.g., MET) and opioid-receptor antagonists (e.g., naloxone) could be used as antitumor drugs. Thus, it is vital to ascertain the opioid receptors/signaling pathways involved in the proliferative/antiproliferative effects mediated by opioids on cancer cells. More in vitro and in vivo studies must be developed to confirm the promising antitumor-mediated impacts of opioid peptides and opioid-peptide antagonists. This research line must be urgently developed and potentiated.

## 5. Future Research

According to previous sections, there is much to do regarding the involvement of the opioid system in cancer. Some of the research lines that would be developed/improved are suggested in this section.

### 5.1. Basic Research

In general, the involvement of MET in cancer has been studied more than the involvement of LEU, beta-END, and DYN. Much information is currently lacking regarding the roles played by the last three peptides in tumor progression. For example, in some tumors (e.g., brain, breast, cervical, cutaneous squamous cell carcinoma, head and neck, larynx, renal, retinoblastoma, testicular, thymic), it is currently unknown whether specific opioid peptides exert or not proliferative/antiproliferative actions and even in some tumors the expression of peptides belonging to the opioid-peptide family is unknown. Opioid peptides (e.g., beta-END) have been reported in tumors, but it is currently unknown whether the peptide favor or not a proliferative/antiproliferative action on tumor cells [140]. MET has been reported in human retinoblastoma [282] and Beta-END-binding sites in the human retinoblastoma McA and Y79 cell lines [283]. However, the physiological significance of these findings is currently unknown. Thus, basic information is lacking; this crucial knowledge must be urgently tackled to develop antitumor strategies.

### 5.2. MET/LEU Research

Most of the studies on the involvement of enkephalins in cancer have been focused on MET; thus, the role played by LEU in cancer progression is practically unknown, meaning that the work that remains to be done is enormous. A higher MET level has been related to a lower brain tumor degree [134], and MET-binding sites decrease with increasing malignancy of gliomas [136]. MET favors apoptotic mechanisms in rat C6 glioma cells [135], and it seems that the inactivation of the MET/opioid receptor system inhibits the blockade action exerted by MET on the growth of astrocytes, promoting glioma development. Previous data open the door to a promising anti-glioma strategy using MET as an antitumor agent; however, this must be studied in-depth since MET also favored the growth of human U-373 MG astrocytoma cells [139]. This note highlights the importance of identifying the opioid receptors/signaling pathways involved in tumor proliferative and antiproliferative mechanisms. An important finding is known in this sense: a shift from mu-opioid receptors in low-grade gliomas to delta-opioid receptors in high-grade gliomas occurs [136].

MET blocked the proliferation of hepatocellular carcinoma cells, and silencing the opioid growth factor receptor favored their proliferation [199]. In metastasis-positive human livers, the level of MET was higher than that found in normal livers [165]. Therefore, MET increase could be an endogenous mechanism to counteract tumor progression. This hypothesis must be confirmed. Moreover, two patients with hepatoblastoma were cured after surgical resection and treatment with MET and naltrexone (low dose) [200]. This promising finding must be confirmed in a higher number of patients.

Other findings must be studied in-depth and confirmed. For example, a lipid conjugation of MET increased tumor suppression activity against human pancreatic adenocarcinomas [250]. This result must be confirmed in other tumors. Mu-opioid receptors mediate head and neck squamous cell carcinoma growth by activating the proliferation/migration of cancer cells [179]. Downregulation of the opioid growth factor receptor favored the progression of head and neck squamous cell carcinoma [183]. These effects must be investigated in-depth. The MET level was increased in prolactin-releasing human pituitary adenoma [265]; however, its physiological significance is unknown. Finally, LEU reduced the invasive capacity of murine colon 26-L5 adenocarcinoma cells [166]; this must be investigated in human colorectal cancer cells.

### 5.3. Beta-END Research

Although essential data are known (see Table 2 and Figure 11), much remains to be done to understand the role of beta-END in cancer fully. This peptide activates in vitro the survival/mitogenic signaling pathways in human breast cancer cells (MDA-MB-231) [149]; however, a correlation between an increased plasma beta-END level and an improved quality of life in patients with breast cancer has been reported [151]. Beta-END seems to counteract breast cancer development by favoring the immune-mediated anticancer defenses in vivo [156,157]. Another experiment has shown that ultraviolet A eye irradiation reduced colon carcinoma symptoms; however, these beneficial effects were reduced when beta-END inhibitors were administered and totally disappeared after the administration of naltrexone [168]. However, the mechanisms by which beta-END exerts these beneficial effects are currently unknown.

Moreover, data suggest that hypothalamic beta-END neuronal transplants blocked colon cancer progression by decreasing inflammatory mechanisms and the epithelial-mesenchymal transition [169]. This research line must be developed because animals with transplanted beta-END neurons showed a lesser adenoma development in the colon, and beta-END-transplanted neurons also prevented hepatocellular carcinoma formation [169,201]. Rats with transplanted neurons expressing beta-END into the hypothalamic paraventricular nucleus showed a protective effect against prostate cancer development, increased immune response, and counteracted inflammatory mechanisms [276]. Other experiments showed that beta-END administration into the raphe magnus nucleus facilitated metastasis inhibited with naloxone; however, metastasis was significantly attenuated when the nucleus was electrically stimulated [170]. The molecular mechanisms involved in these processes must be elucidated.

### 5.4. DYN Research

After LEU, DYN is the less-known reviewed peptide concerning its implication in cancer progression. Like LEU, the work that remains for DYN is enormous. U50,488, a kappa-opioid receptor agonist, favored apoptosis in human multiple-myeloma cells expressing mu- and kappa-opioid receptors; but the antiproliferative effect was not mediated by kappa receptors [235]. This result demands further investigation. DYN A_1-17_ is the major component observed in pheochromocytomas, and DYN A_1-13_ and DYN A_1-12_ are minor components, but it is currently unknown what the physiological significance of these findings is [263,264]. The DYN A_2-13_ fragment failed to increase tumor cell growth, but DYN A, DYN A_1-13_, and DYN A_1-7_ favored the growth of the prostatic carcinoma cell line (DU145); naloxone inhibited the effect mediated by DYN A but increased the growth of tumor cells [277]. This study is vital because fragments of the opioid peptides mediate different results; this issue must be fully developed using opioid fragments in other tumors to know their proliferative/antiproliferative actions. These actions could be mediated by different proteins G/receptors but await in-depth elucidation.

Regarding DYN A, other research lines must be investigated in-depth. For example, DYN favored apoptosis in neuroblastoma cells, but the anesthetic isoflurane inhibited this effect [242]. At high concentrations, DYN A binds to neuropeptide Y receptors (Y1 and Y2) in neuroblastoma cells; however, the physiological significance of this observation is unknown [243].

### 5.5. Signaling Pathways

Some data regarding the opioid-mediated signaling pathways involved in cancer (see Figure 9) should be analyzed. For example, the MET/opioid growth factor receptor system augmented the cyclin-dependent kinase inhibitor p21 protein expression, decreasing the progression of human pancreatic cancer [251]. However, the signaling pathways that mediate the proliferative or antiproliferative mechanisms controlled by opioids in tumor cells deserve more investigation. Determining the signaling pathways leading to cancer triggered by opioids is a promising research line that must be fully developed.

### 5.6. Tumor Cell Migration

Tumor cell migration is a crucial issue that must be studied in-depth to develop future antitumor strategies. The use of opioid peptide receptor antagonists as antimetastatic agents must be exploited since opioid peptides (e.g., MET) promote the migration of cancer cells (e.g., breast cancer cells) [145]. In contrast, using opioid peptides (e.g., LEU) is also a promising antimetastatic strategy because they reduce cancer cells’ invasive capacity [166]. Currently, it is not known whether opioid peptides favor or not tumor cell migration in many cancer types.

### 5.7. Angiogenesis

*Pro-opiomelanocortin* gene delivery inhibited the vasculature of tumors and blocked the growth of lung carcinoma cells by apoptotic mechanisms and melanoma progression and metastasis [219]. It is an important finding, but much information is lacking regarding the involvement of opioid peptides in tumor angiogenesis. This information is essential to develop anti-angiogenic strategies.

### 5.8. Polymorphisms

Another topic that must be studied in-depth is the expression of polymorphisms in patients with cancer. In this sense, a study has reported that patients with a polymorphism in the mu-opioid receptor gene showed better survival [143]. This research must be fully developed.

### 5.9. Substances Regulating the Opioid System

MET upregulates the expression of opioid receptors [175]; imiquimod also upregulates this expression, increasing the antitumor effect mediated by MET [225], and the downregulation of opioid receptors favors the progression of tumors [183]. Nicotine partially or totally reversed opioid-induced growth suppression in lung cancer cells [208]. These cells express mu, kappa, and delta opioid receptors, and their agonists block the growth of tumor cells [208]. Nicotine favored the release of DYN and alpha-neo-END peptides from pheochromocytomas [262]. Cinobufagin, used for the treatment of cancer pain, relieved cancer pain by upregulating the expressions of both mu-opioid receptors and beta-END in the hind paw tumor and tissues placed close to the tumor in a rat model of paw bone cancer pain [131]. The knowledge of the different compounds that regulate the release of opioid peptides is essential since, for example, DYN favors apoptosis in neuroblastoma cells but also promotes the growth of prostate cancer cells (see Table 3). Knowing how endogenous or exogenous substances regulate the release of opioid peptides from nerve terminals or from normal/tumor cells is crucial to establishing anticancer strategies.

### 5.10. Endopeptidases

A released dibasic cleaving peptidase that converts DYNs (e.g., DYN A_1-12_, DYN A_1-9_, proDYN B) into LEU-Arg^6^ has been obtained from the medium of EL-4 mouse thymoma cells [288]. Much information is lacking regarding the involvement of opioid endopeptidases in tumor progression.

### 5.11. The Opioid System as a Cancer-Predictive Factor

This topic must also be developed, and the published promising data confirmed: an increased risk for developing breast cancer has been associated with a low plasma level of pro-enkephalin [146]; beta-END immunoreactivity was lower in benign melanocytic naevi than in metastatic and advanced melanomas [232], and a positive correlation between plasma beta-END levels and survival time/disease-free time has been suggested in patients with ovarian cancer: lower beta-END concentrations were observed in patients with recurrence than in those without recurrence [248]. MET improved survival/clinical symptoms in patients with advanced pancreatic cancer [252]; high MET plasma levels in patients with pancreatic cancer have been reported [253]; zeta-opioid receptors are more expressed in prostate cancer tissues than in normal ones, and this expression was high in aggressive and undifferentiated prostate cancer tissues [272]. An association has been reported between high mu-opioid receptor expression and poorer survival in patients with prostate cancer [273] and a correlation between increased plasma beta-END levels and improved quality of life in patients with breast cancer [151].

### 5.12. Structural Dynamics of OR

The analysis of the structural dynamics, conformational states, and functional splice variants of OR related to cancer development and progression deserves further investigation to ascertain better their influence on molecular events that initiate abnormal cell proliferation in different tissues [295]. One aspect calling for specific attention is the definition of OR biased and cross-talk signaling events that may force the activity of OR toward activating metabolic routes firing cancer appearance and progression [113,296,297]. The study of structural and biochemical mechanisms should be accompanied by population observations relating to the influence of opioid systems in different cancers, considering putative-associated or confounding factors [298,299].

### 5.13. Other Research Lines

Some studies have suggested microbiota involvement in cancer, especially gastrointestinal cancer [300]. In this sense, it has been stated that a dysregulation of the expression of non-coding RNA through the gut microbiome could be the origin of gastrointestinal cancer (see [300] for review). This is an exciting line of research that must be investigated and developed.

Although it is known that peptides derived from proopiomelanocortin, including ACTH, regulate cancer cell proliferation [301], the effect of ACTH on opioid peptides in cancer development is currently unknown.

Opioid receptors might control apoptotic mechanisms in cancer cells by modulating apoptosis-related proteins, including tumor-suppressor protein p53. For example, an endomorphin 2 analog reduced the expression of Bcl-2 and increased the expression of Bax, p53, and pro-caspases 3 and 9 in human DLD-1 and RKO colon cancer cells [302].

## 6. Conclusions

MET is the opioid peptide more studied in cancer, followed by beta-END. Most of the studies performed on opioid peptides are descriptive since they were mainly focused on the expression or not of these peptides and their receptors in tumor cells. In many cancer types, fragmentary, scarce, or absent data are currently available regarding the involvement of these peptides in cancer development. This fact applies to DYN and LEU. Thus, a systematic study on the expression/role (antiproliferative/proliferative effects, metastasis, angiogenesis) of opioid peptides and their receptors in many types of cancers is urgently needed. This study must include the involvement of peptide fragments in tumor progression. It is also important to know whether the expression of opioid receptors is essential for the viability of tumor cells and whether tumor cells express more opioid receptors than normal cells. Secondly, more studies must focus on using opioid peptides (e.g., MET) and opioid-receptor antagonists/agonists as antitumor agents. In particular, an anticancer effect is generally mediated by MET against different tumors, although the peptide also promotes the growth of some tumors. Thus, depending on the type of cancer, opioid-peptide agonists or antagonists could be used as antitumor agents. It is crucial to know in-depth which opioid receptors and signaling pathways are involved in cancer progression to establish better antitumor strategies. In the same way, the use of opioid-receptor antagonists must be fully developed since only a few studies have been focused on the anticancer actions mediated by these antagonists; in addition, the antitumor effect mediated by other antagonists than those currently studied must be tested in pre-clinical studies. Developing specific opioid-receptor antagonists exerting an antitumor action is crucial without affecting other receptors. Consequently, the structure-function relationships between opioid peptides and their receptors must be well known to design new broad-spectrum antitumor drugs and for drug-design studies.

More studies must be focused on the tumor microenvironment. For example, the cross-talk between cancer development and the release of opioid peptides from nerve fibers must be studied in-depth because these peptides control the progression of tumors. The knowledge about the agents that control the expression of opioid peptides and their receptors must be increased to develop new antitumor strategies. Peptides regulate the immune response in cancer; this is important since, for example, beta-END favors the immune-mediated anticancer defenses [156,157], MET increases immunogenicity and recognition of cancer cells, blocks the formation of inflammatory cytokines and decreases immune checkpoints in colorectal cancer cells, and MET reduced immunosuppression in cutaneous squamous cell carcinoma. It must also be investigated how DYN A interacts with other surface receptors since this peptide binds to neuropeptide Y receptors (Y1 and Y2) in neuroblastoma cells and the effects mediated by opioid peptides regulating the glycolysis in tumor cells because this process is essential for tumor cells to obtain ATP. This knowledge will lead to developing future new anticancer strategies. Combined antitumor methods using peptides/opioid-receptor antagonists and chemotherapy must be performed, and treatments increasing the chemosensitivity of cancer cells must be tested.

Moreover, new tracers such as radiopharmaceuticals (e.g., opioid peptide-drug conjugates) for cancer imaging and treatment is a promising and attractive research field that must be developed. Moreover, opioid peptides/receptors as prognostic factors or tumor biomarkers must be studied in-depth (e.g., higher MET level: lower tumor degree; increasing glioma malignancy: MET-binding sites decrease; beta-END level and tumor progression). Also, the confirmation of their role in several tumors and studies focused on single nucleotide polymorphisms.

Although the involvement of peptides in cancer is a promising line of research, there is much to investigate regarding the participation of opioid peptides in cancer progression: basic information is scarce, incomplete, or absent in many tumors, and, in fact, no clinical study regarding the antitumor action of opioid peptides/opioid-peptide receptor antagonists has been developed to date. According to the current and future knowledge acquired, promising anticancer strategies could be developed alone or in combination therapy with chemotherapy/radiotherapy.

## Figures and Tables

**Figure 1 biomedicines-11-01993-f001:**
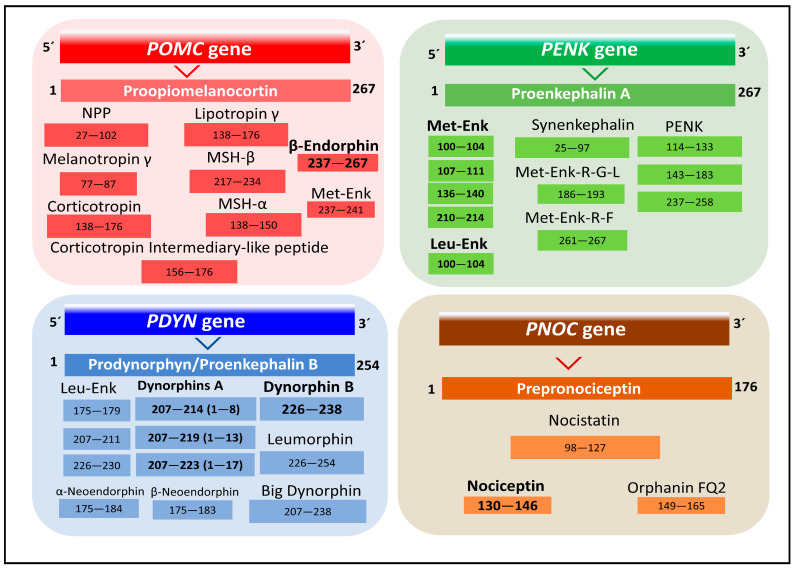
Representation of the four families of the endogenous opioid peptides. The numbers indicate amino acid positions in the respective peptide precursor (UniProt codes P01189, Proopiomelanocortin; P01210, Proenkephalin A; P01213, Prodynorphin/Proenkephalin B; Q13519, Prepronociceptin, [19]).

**Figure 2 biomedicines-11-01993-f002:**
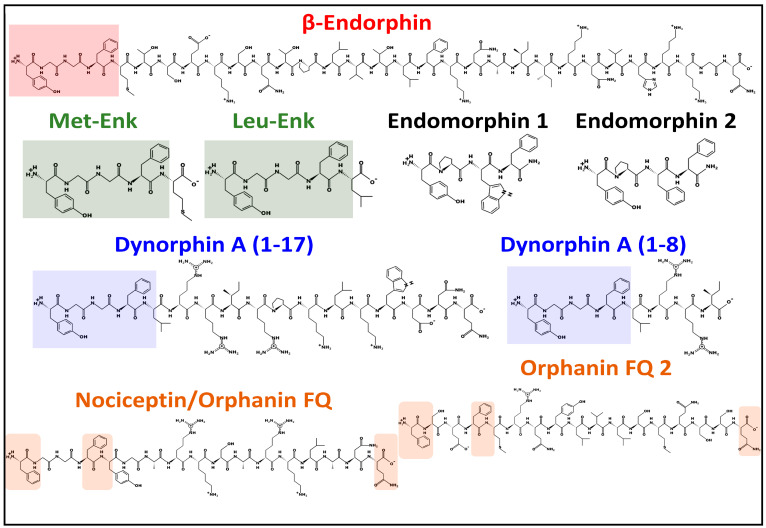
Structure of representative endogenous opioids. Colored squares highlight the shared N-terminal sequence, YGGF, in β-endorphin, enkephalins, dynorphin A (1–17), and dynorphin A (1–8). Colored squares in nociceptin and orphanin depict common amino acid positions. All peptides shown have a Phe residue at the fourth position. All peptides shown, except heptadecapeptides, nociceptin, and orphanin FQ2 (sharing a Phe residue), share a Tyr residue at the N-terminus. All peptide structures were drawn at pH = 7 with PepDraw [29], the free web-based software from Tulane University (https://pepdraw.com/ (accessed on 17 May 2023)).

**Figure 3 biomedicines-11-01993-f003:**
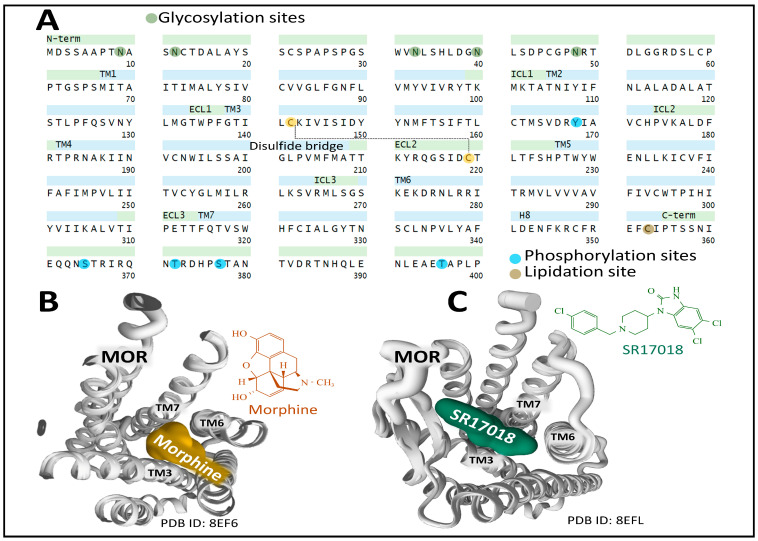
Primary structure and helices organization of the MOP receptor, depicting posttranslational modifications (**A**) [19,49,50]. An external view from the extracellular site of the structure of human MOP bound to the agonists morphine (**B**) and SR17018 (5,6-dichloro-3-[1-[(4-chlorophenyl)methyl]piperidin-4-yl]-1H-benzimidazol-2-one) (**C**) shows the interaction of both agonists with transmembrane domains 3, 6, and 7. Figures were obtained from the Protein Data Bank [51] and correspond to PDB ID 8EF6 and 8EFL [52] and drawn with the free web-based open-source toolkit Molstar (https://molstar.org/ (accessed on 17 May 2023) [53]. Two-dimensional structures of morphine and SR17018 were drawn with KingDraw software [54] (version 1.1.0) (https://www.kingdraw.cn/en/ (accessed on 17 May 2023)).

**Figure 4 biomedicines-11-01993-f004:**
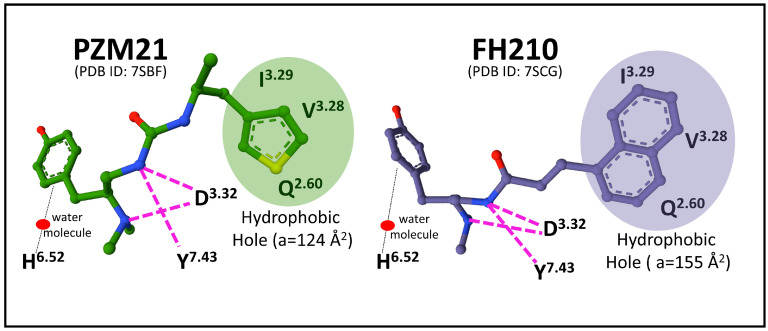
Representation of the main interactions of MOP-biased agonists PZM21 (1-[(2S)-2-(dimethylamino)-3-(4-hydroxyphenyl)propyl]-3-[(2S)-1-thiophen-3-ylpropan-2-yl]urea) and FH210 (a naphthyl-substituted acryl amide derivative of PZM21) with MOP-Gi complex (see text for details). The structures were obtained from the Protein Data Bank (PDB ID 7SBF and 7SCG, [62]) and drawn with the free web-based open-source toolkit Molstar (https://molstar.org/ (accessed on 17 May 2023)) [53]). Amino acid positions are numbered according to Ballesteros and Weinstein [57].

**Figure 5 biomedicines-11-01993-f005:**
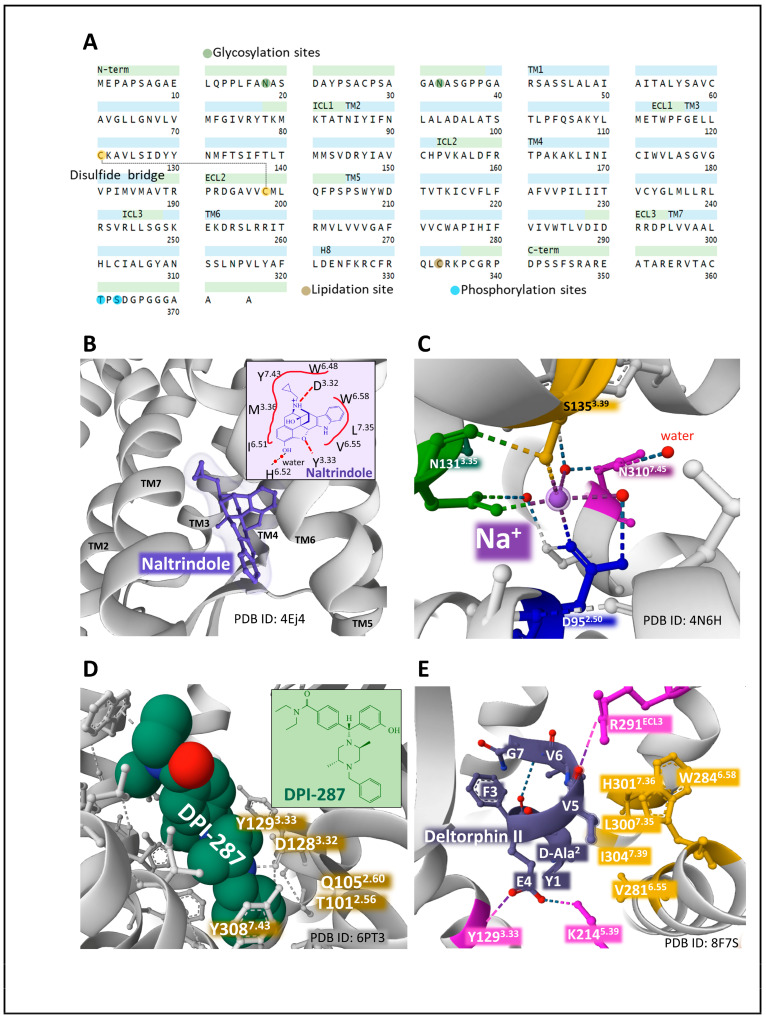
The primary structure and domains of the DOP receptor with the indication of posttranslational modifications (**A**) [19,49,50]. The structure of murine DOP bound to the antagonist naltrindole (**B**) and the 2D structure of naltrindole depicting close interactions with DOP amino acid residues (**B**, inset). (**C**) represents the allosteric sodium site of human DOP and the contacts established with residues of the receptor. (**D**) shows the pocket of human DOP for the agonist DPI-287 and the interactions with DOP residues (inset). E highlights the interaction of peptide agonist deltorphin II with human MOP. The interactions of the amino acid residues of the peptide with the receptors illustrate the orthosteric binding cavity (**E**). Figures obtained from the Protein Data Bank [51] correspond to PDB ID 4EJ4 [64], 4N6H [65], 6PT3 [66], and 8F7S [67] drawn with the free web-based open-source toolkit Molstar (https://molstar.org/ (accessed on 17 May 2023)) [53]. Two-dimensional structures of naltrindole and DPI-287 were drawn with KingDraw software (version 1.1.0) (https://www.kingdraw.cn/en/ (accessed on 17 May 2023)). Amino acid residues are numbered according to Ballesteros and Weinstein’s nomenclature [57].

**Figure 6 biomedicines-11-01993-f006:**
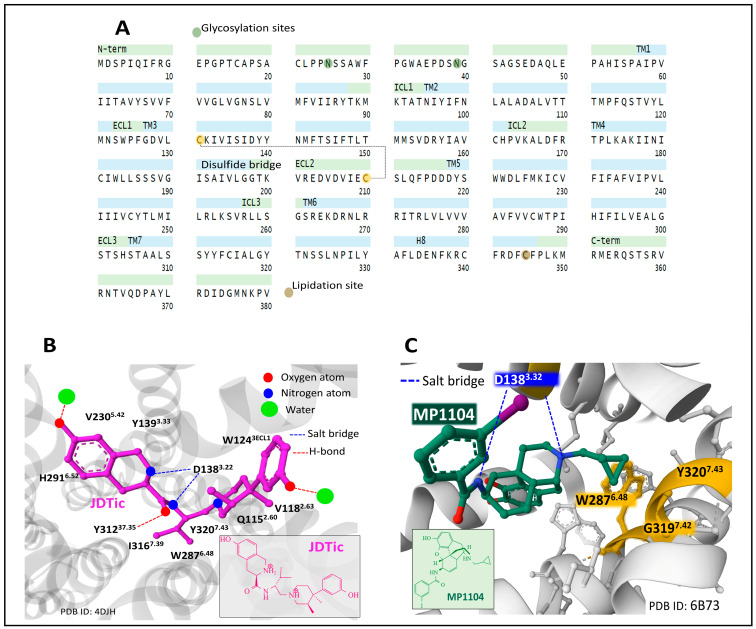
The primary structure and domains of the KOP receptor with the indication of posttranslational modifications (**A**) [19,49,50]. The structure of human KOP is bound to the selective antagonist JDTic ((3R)-7-hydroxy-N-[(2S)-1-[(3R,4R)-4-(3-hydroxyphenyl)-3,4-dimethylpiperidin-1-yl]-3-methylbutan-2-yl]-1,2,3,4-tetrahydroisoquinoline-3-carboxamide) (2D structure in the inset), indicating some amino acid contacts which secure its position within the orthosteric cave (**B**). The binding of non-selective agonist MP1104 (N-[(4R,7R,7aR,12bS)-3-(cyclopropylmethyl)-9-hydroxy-2,4,4a,7,7a,13-hexahydro-1H-4,12-methanobenzofuro[3,2-e]isoquinolin-7-yl]-3-iodobenzamide) (2D structure in the inset) to KOP reveals the ionic interaction of the ligand with D138^3.32^ and the hydrophobic hole delimited by W287^6.48^, G319^7.42^, and Y320^7.43^ (**C**). Figures obtained from the Protein Data Bank [51] correspond to PDB ID 4DJH [71] and 6B73 [72], drawn with the free web-based open-source toolkit Molstar (https://molstar.org/ (accessed on 17 May 2023)) [53]. Two-dimensional structures of JDTic and MP1104 were drawn with KingDraw software (version 1.1.0) (https://www.kingdraw.cn/en/ (accessed on 17 May 2023)). Amino acid residues are numbered according to Ballesteros and Weinstein’s nomenclature [57].

**Figure 7 biomedicines-11-01993-f007:**
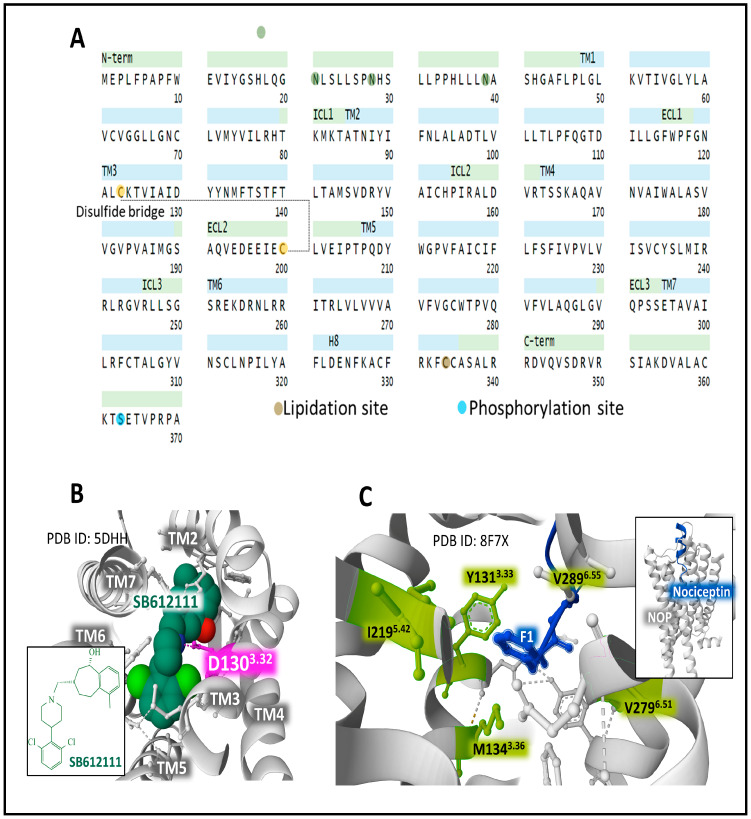
The primary structure and domains of the NOP receptor with the indication of posttranslational modifications (**A**) [19,49,50]. The structure of human NOP is bound to the selective antagonist SB612111 ((5S,7S)-7-[[4-(2,6-Dichlorophenyl)piperidin-1-yl]methyl]-1-methyl-6,7,8,9-tetrahydro-5H-benzo [7]annulen-5-ol)) (2D structure in the inset), indicating amino acid contacts in the orthosteric binding pocket (**B**). The binding nociceptin to NOP (**C**) reveals the hydrophobic contacts of the ligand with the inferior domain of the binding site delimited by Y131^3.33^, M134^3.36^, I219^5.42^, V279^6.51^, and V283^6.55^. The inset figure in C illustrates the position of nociceptin within the orthosteric site of NOP. Figures are from the Protein Data Bank [51] formatting and correspond to PDB ID 5DDH [80] and 8F7X [67], drawn with the free web-based open-source toolkit Molstar (https://molstar.org/ (accessed on 17 May 2023)) [53]. The two-dimensional structure of SB612111 was drawn with KingDraw software (version 1.1.0) (https://www.kingdraw.cn/en/ (accessed on 17 May 2023)). Amino acid residues are numbered according to Ballesteros and Weinstein’s nomenclature [57].

**Figure 8 biomedicines-11-01993-f008:**
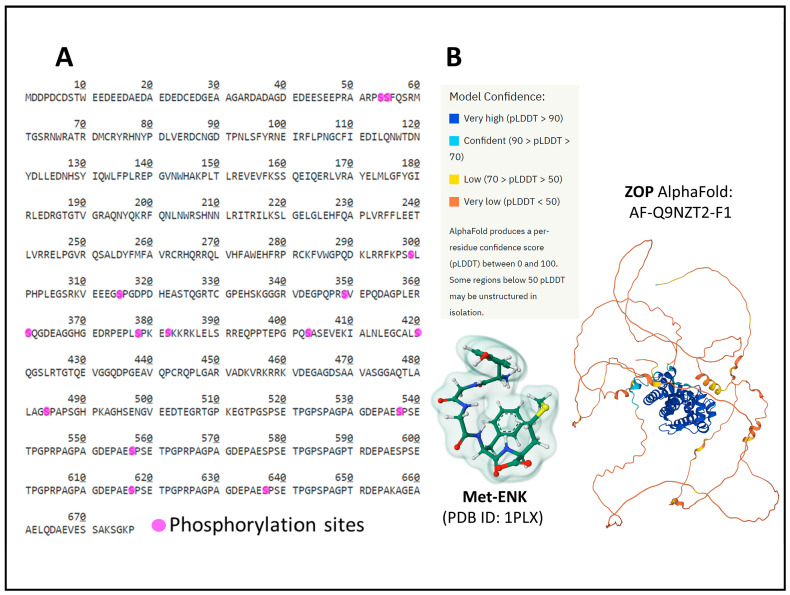
The primary structure of the ZOP receptor with the indication of posttranslational phosphorylation sites (**A**) [19]. Panel (**B**) represents the AlphaFold prediction [87,88] of the human ZOP structure and the structure (Gaussian volume) of its ligand Met-ENK. The figure of Met-ENK is from the Protein Data Bank [51] and corresponds to PDB ID 1PLX [89], drawn with the free web-based open-source toolkit Molstar (https://molstar.org/ (accessed on 17 May 2023) [53]). The AlphaFold representation is from the AlphaFold Protein Structure Database (https://alphafold.ebi.ac.uk/entry/Q9NZT2, accessed on 12 June 2023)).

**Figure 9 biomedicines-11-01993-f009:**
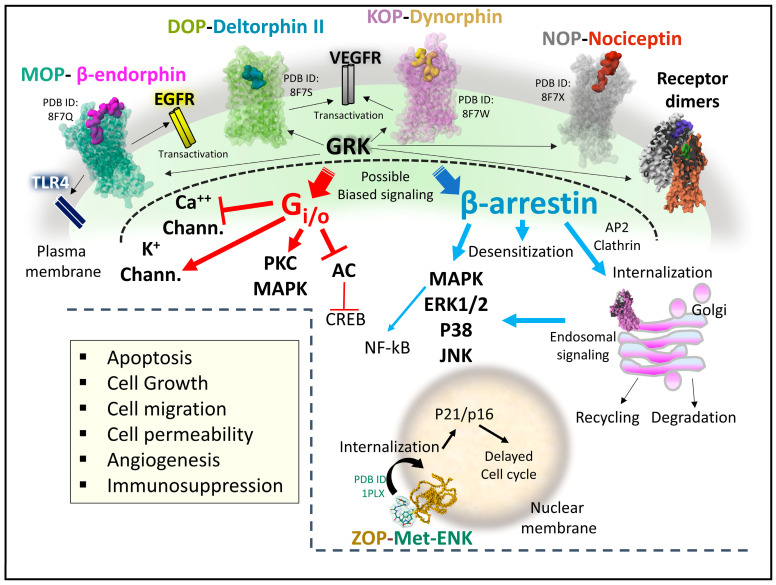
Representative signaling pathways relating opioid receptors activity and cancer development. OR structures 87FQ (MOP-beta-endorphin), 87FS (DOP-deltorphin II, 87FW (KOP-dynorphin, 87FX (NOP-nociceptin) [67] and 1PLX (Met-ENK structure, Gaussian volume, [89], are from the Protein Data Bank [51]. The AlphaFold representation of ZOP is from the AlphaFold Protein Structure Database (https://alphafold.ebi.ac.uk/entry/Q9NZT2, accessed on June 12, 2023)). All receptor structures were drawn with the free web-based open-source toolkit Molstar (https://molstar.org/ (accessed on 17 May 2023) [53]). Abbreviations: AC, adenylyl cyclase; CREB, cAMP response element-binding protein; EGFR, epidermal growth factor receptor; ERK, extracellular receptor kinase; GRK, G-protein-coupled receptor kinase; JNK, c-Jun N-terminal kinase; MAPK, mitogen-activated protein kinase; NF-kB, Nuclear factor kappa-light-chain-enhancer of activated B cells; PKC, protein kinase C; VEGFR, vascular endothelial growth factor receptor; TLR4, toll-like receptor 4.

**Figure 10 biomedicines-11-01993-f010:**
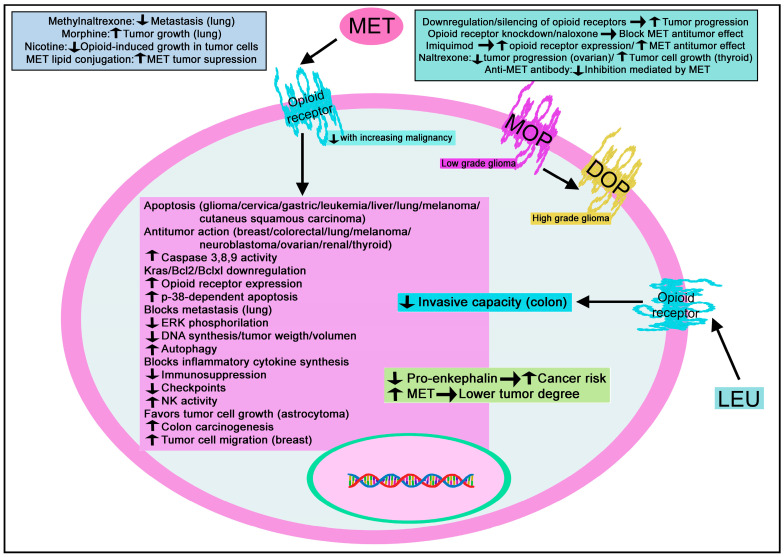
Summary of the mechanisms mediated by MET/LEU in cancer. DOP: delta-opioid receptor; ERK: extracellular signal-regulated kinase; MOP: mu-opioid receptor; NK: natural killer cells. ↑: increase; ↓: decrease.

**Figure 11 biomedicines-11-01993-f011:**
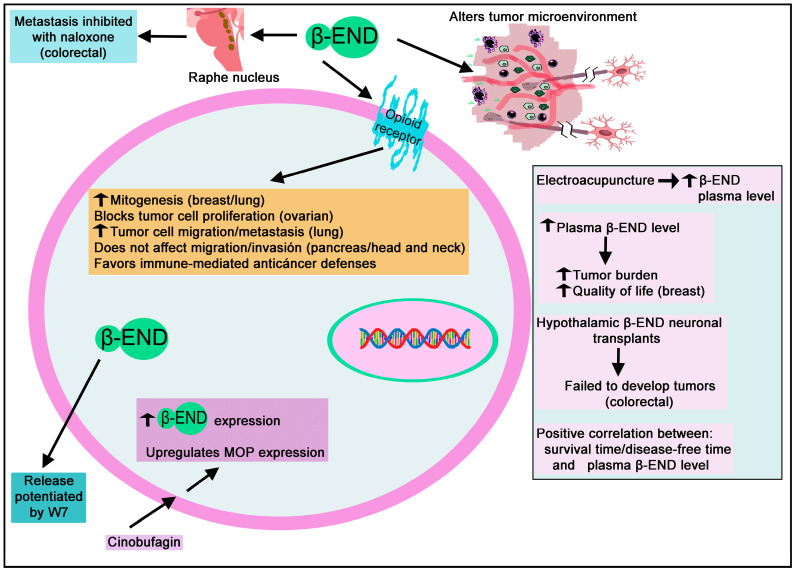
Summary of the mechanisms mediated by beta-END in cancer. ↑: increase; ↓: decrease.

**Figure 12 biomedicines-11-01993-f012:**
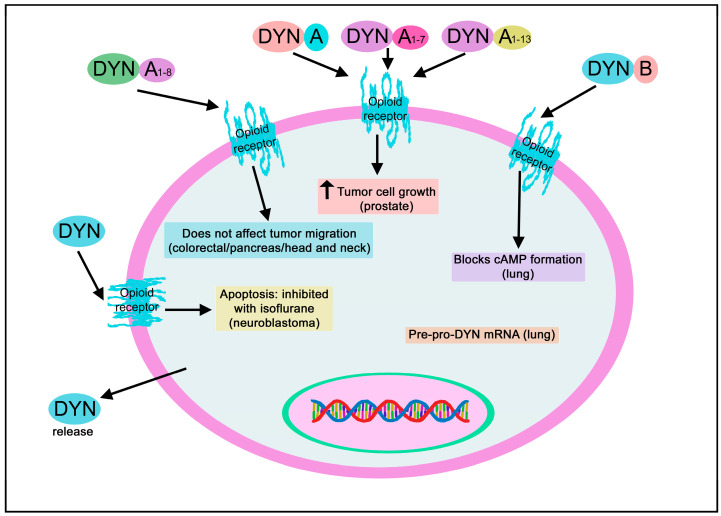
Summary of the mechanisms mediated by DYN in cancer. cAMP: cyclic adenosine 3′, 5′-monophosphate.

**Table 1 biomedicines-11-01993-t001:** Involvement of MET/LEU in cancer.

Tumor	MET/LEU	References
Brain	Pro-enkephalin/MET in human tumors. Higher MET level, lower tumor degree. MET promotes apoptosis in rat C6 glioma cells and increases the activity of caspases 3, 8, and 9. MET-binding sites decrease with increasing malignancy of gliomas. A shift from mu-opioid receptors in low-grade gliomas to delta-opioid receptors in high-grade gliomas occurs. MET favors the growth of human U-373 MG astrocytoma cells.	[134,135,136,139]
Breast	MET/LEU expression in cells and stroma of human cancer samples. MET, but not LEU, stimulated the migration of MDA-MB-468 human carcinoma cells. A low-fasting plasma level of pro-enkephalin correlated with an increased risk of cancer development in postmenopausal/middle-aged women. MET/opioid growth factor receptor system blocked the proliferation of triple-negative breast cancer cell lines.	[144,145,146,147]
Cervical	MET blocked cervical carcinoma progression and induced apoptosis. MET decreased the number of myeloid-derived suppressor cells in both tumor and circulation.	[159]
Colorectal	Enkephalins in rectal carcinoids. MET/LEU did not affect tumor cell migration, chemotaxis, or invasion. MET did not alter the viability of tumor cells. MET enhanced colon carcinogenesis. MET exerts an antitumor action against colorectal tumors (MC38 cell line). MET augmented immunogenicity and recognition of tumor cells; downregulated Kras, Bcl2, and Bclxl; blocked the synthesis of inflammatory cytokines; and reduced immune checkpoints in tumor cells. Naltrexone blocked the antitumor action induced by MET. CD10 increased colorectal cancer cell metastasis by abrogating the antitumor action mediated by MET. MET decreased the phosphorylation of ERK/epidermal growth factor receptors and increased p38-dependent apoptosis. LEU decreased the invasive capacity of murine colon 26-L5 adenocarcinoma cells.	[162,163,164,165,166,172,186]
Cutaneous squamous cell carcinoma	MET blocked the cell growth of tumor cells by apoptosis. MET blocked the proliferation of A431 cells by apoptosis and promoted autophagy in tumor cells. MET decreased immunosuppression by reducing the number of myeloid-derived suppressor cells.	[173,206]
Gastric	MET blocked the growth of human tumor cells by apoptosis. MET upregulated the expression of the opioid growth factor receptor. Tumor cell apoptosis, mediated by MET, was blocked when the opioid receptor expression was in knockdown.	[175,176]
Head and neck	MET expression in human head and neck squamous cell carcinoma. MET/LEU did not affect tumor cell migration, chemotaxis, or invasion. MET did not affect either the differentiation or the viability of tumor cells. Reduced DNA synthesis and tumor weight/volume after treatment with MET. Downregulation of the opioid growth factor receptor favored tumor progression. LEU in head and neck paragangliomas.	[172,180,181,183,184,186,187]
Larynx	MET in neuroendocrime tumors (paragangliomas).	[188]
Leukemia	Pro-ENK A in leukocytes from patients with chronic lymphoblastic leukemia. MET inhibited the metabolic activity of the leukemic NALM-1 cell line. MET promoted apoptosis in K562 human erythroid leukemia cells. MET favored pre-B acute lymphoblastoid cell migration.	[190,191,192,193]
Liver	MET concentration was higher in metastasis-positive human livers than in normal livers. Proliferation of hepatocellular carcinoma cells: inhibited after treatment with MET. Silencing of the opioid growth factor receptor promoted the proliferation of hepatocellular carcinoma cells. Patients with hepatoblastoma are cured after surgical resection and treatment with naltrexone and MET.	[165,199,200]
Lung	Pro-enkephalin/MET expression. Serum LEU levels are higher in patients with bronchial carcinoma than in control individuals. MET exerted an antitumor effect against cancer cells: this was abolished in growth factor receptor knockdown. MET increased the expression of opioid growth factor receptors and promoted apoptosis in cancer cells. Methylnaltrexone counteracted Lewis lung carcinoma growth and decreased metastasis; morphine favored tumor growth. Nicotine partially/totally reversed opioid-induced growth suppression in cancer cells. MET blocked pulmonary metastasis and enhanced the activity of NK cells.	[202,203,205,206,207,208,209]
Melanoma	MET/LEU expressions decreased in melanocytic tumors. MET exerted an antitumor effect in mice xenografted with B16-BL6 cells; this was inhibited with naloxone. Tumor growth/cell dissemination was counteracted with MET; the peptide blocked A375 cell proliferation through apoptosis. Imiquimod upregulated the expression of the opioid growth factor receptor, increasing the MET antitumor effect. MET promotes cell cycle arrest and increases the expression of opioid receptors in B16 cells. MET decreased tumor weight/volume. MET promoted cell cycle arrest and favored apoptosis in human A375 cells.	[220,222,223,224,225]
Neuroblastoma	MET in mouse Neuro2a cells. MET arrested the growth of human SK-N-SH cells. MET decreased the tumor mitotic index, which was counteracted with naltrexone.	[236,237]
Ovarian	Enkephalins in ovarian carcinoids. MET/opioid growth factor receptor in human cancer cells. MET exerted an inhibitory proliferative effect on tumor cells. MET neutralization: cell proliferation and silencing of the opioid growth factor receptor favored tumor cell replication. MET delayed cells moving. Naltrexone inhibited tumor progression and, with cisplatin, exerted an enhanced inhibitory effect. MET blocked the proliferation of human ovarian KF cancer cells; this effect was counteracted with naloxone.	[16,244,245,246,249]
Pancreatic	High MET plasma level in patients with pancreatic cancer. MET/LEU did not affect the migration, chemotaxis, or invasion of tumor cells (MIA PaCa-2, PANC-1, BxPC-3) MET lipid conjugation: increased MET tumor-suppression activity in human pancreatic adenocarcinomas. MET/opioid growth factor receptor system attenuated tumor progression. MET ameliorated clinical symptoms/survival in patients with advanced pancreatic cancer.	[186,250,251,252,253]
Pheochromocytoma	MET/LEU, proprotein convertase 2 and pro-enkephalin in human pheochromocytomas. Nicotine promoted MET release from pheochromocytoma cells. MET/LEU ratio is higher in extramedullary than in medullary pheochromocytomas.	[255,256,257,258]
Pituitary	MET level increased in prolactin-releasing human pituitary adenomas.	[265]
Prostate	DAGO/DSLET blocked the proliferation of tumor cells; this was counteracted with diprenorphine. LEU in prostatic carcinomas.	[274,275]
Renal	MET blocked the proliferation of human cancer cells.	[281]
Retinoblastoma	MET expression in humans.	[282]
Thymic	MET in a thymic carcinoid.	[285]
Thyroid	Human anaplastic thyroid cancer cells: MET/opioid growth factor receptor expression. MET blocked cell replication. Naltrexone promoted tumor cell growth, and anti-MET antibodies counteracted the inhibitory action mediated by MET.	[289]

**Table 2 biomedicines-11-01993-t002:** Involvement of beta-END in cancer.

Tumor	Beta-Endorphin	References
Bone	Cinobufagin promoted beta-END mRNA and protein expressions in microglia. It upregulated the expressions of mu-opioid receptors and beta-END in the tumor and tissues placed close to the tumor.	[130,131]
Brain	Beta-END-binding sites in human glioblastoma cells. Beta-END in human brain tumor cyst fluids.	[140,141]
Breast	High plasma beta-END levels in healthy women were even higher in healthy postmenopausal women; lower in women with breast cancer. Chemotherapy improved beta-END levels in postmenopausal women. END did not affect the migratory capacity of human breast carcinoma cells. Beta-END expression in cells and stroma of human breast cancer samples. Beta-END activates survival/mitogenic signaling pathways in human cancer cells. Increasing plasma beta-END levels correlated with increasing tumor burden. A correlation between increased plasma beta-END levels and improved quality of life in breast cancer patients. Beta-END counteracts breast cancer development by favoring immune-mediated anticancer defenses. Beta-END alters the tumor microenvironment.	[144,145,149,151,155,156,157,177]
Cervical	Electroacupuncture increases plasma beta-END levels in humans.	[160]
Colorectal	Beta-END in adenocarcinomas derived from the colon mucosa; its expression is higher in adenocarcinomas than in the mucosal layer of normal colons. Beta-END in rectal carcinoids. Beta-END did not affect tumor cell migration, viability, chemotaxis, or invasion. Colon carcinoma symptoms decreased after ultraviolet A eye irradiation; these effects were reduced when beta-END inhibitors were administered and totally disappeared with naltrexone. Animals with hypothalamic beta-END neuronal transplants failed to develop tumors and showed a lesser adenoma development. Beta-END administration into the raphe magnus nucleus: favored metastasis inhibited with naloxone, and when this nucleus was electrically stimulated, metastasis was attenuated.	[162,167,168,169,170,172,186]
Gastric	Beta-END in adenocarcinomas derived from the antral mucosa. Plasma beta-END level decreased in patients after transcutaneous electrical stimulation.	[177,178]
Head and neck	Beta-END did not affect the migration, chemotaxis, or invasion of squamous carcinoma cells. Beta-END did not affect either the differentiation or the viability of tumor cells. Beta-END increased the production of leukocyte migration inhibitory factor, reaching almost normal levels in patients with squamous carcinoma.	[172,185,186,187]
Larynx	Beta-END in tumor cells.	[189]
Leukemia	Beta-END in the cerebrospinal fluid of children with acute lymphoblastic leukemia: highest levels observed at the end of the intensification chemotherapy; glucocorticoid treatment decreased beta-END levels. Plasma beta-END levels decreased in patients with solid tumors after treatment with a chemotherapeutic drug. Plasma beta-END levels are higher in patients with acute leukemia than in healthy individuals. Beta-END promoted the growth of T-lymphoblastoid cells.	[194,195,196,197]
Liver	Neurons expressing beta-END transplanted into the hypothalamus prevented hepatocellular carcinoma formation. This strategy inhibited carcinogen-induced liver histopathologies and augmented the concentration of NK cell cytotoxic agents.	[201]
Lung	Beta-END in the bronchoalveolar lavage fluid/plasma of patients. Beta-END in lung small-cell carcinomas and carcinoid tumors. U1,690 cell line expresses beta-END, and the peptide promotes its proliferation through non-opioid binding sites. Beta-END acts as a chemoattractant for small-cell lung carcinoma cells favoring migration and metastasis.	[210,211,212,214,215]
Melanoma	Beta-END in human samples. Beta-END in secondary neuroendocrine carcinomas of the skin but not in primary carcinomas. B16 melanoma cells synthesize and release beta-END. B16 cells decreased tumor growth in mu-opioid receptor-deficient animals. A correlation occurs between beta-END levels and tumor progression. Beta-END immunoreactivity: lower in benign melanocytic naevi than in metastatic and advanced melanomas.	[221,232,233]
Neuroblastoma	Beta-END-binding sites expression. MET decreased the tumor mitotic index, which was counteracted with naltrexone.	[238,239]
Ovarian	Beta-END in ovarian sex cord-stromal tumors and ovarian carcinoids. A positive correlation between patients’ survival time/disease-free time and plasma beta-END levels. Lower beta-END concentrations were observed in patients with recurrence Beta-END blocked the proliferation of human ovarian KF cancer cells; this effect was counteracted with naloxone.	[244,247,248,249]
Pancreatic	Beta-END did not affect tumor cell migration, chemotaxis, or invasion (MIA PaCa-2, PANC-1, BxPC-3).	[186]
Pheochromocytoma	Beta-END expression and release.	[257,259,260]
Pituitary	Beta-END in pituitary adenomas. Beta-END released from human cancer cells. Beta-END presence in clinically silent pituitary corticotroph adenomas. Beta-END/beta-END_1-27_ in extracts of pituitary melanotroph tumors transplanted subcutaneously. W7 potentiated the release of beta-END from cancer cells.	[266,267,268,269,270]
Prostate	Beta-END in prostatic carcinomas. Rats with transplanted neurons expressing beta-END into the hypothalamic paraventricular nucleus showed a protective effect against prostate cancer development.	[275,276]
Retinoblastoma	Beta-END-binding sites in human cell lines.	[283]
Thymic	Beta-END binds to non-opioid binding sites expressed in thymoma cells. Beta-END in an oncocytic carcinoid tumor of the thymus.	[286,287]
Thyroid	Presence of opioid peptides derived from the three opioid precursors in human thyroid medullary carcinomas. Beta-END released from cultured medullary thyroid carcinoma cells.	[11,290]

**Table 3 biomedicines-11-01993-t003:** Involvement of DYN in cancer.

Tumor	DYN	References
Brain	DYN A_1-13_ decreased water content, mediated by glutamate, in C6 glioma cells.	[142]
Breast	DYN A_1-17_/DYN A_1-8_ in Walker 256 tumors.	[158]
Colorectal	DYN A_1-8_ did not affect tumor cell migration, viability, chemotaxis, or invasion.	[172,186]
Head and neck	DYN A_1-8_ did not affect the migration, chemotaxis, or invasion of squamous carcinoma cells. DYN A_1-8_ did not affect tumor cells’ differentiation or viability.	[172,186,187]
Lung	Pre-pro-DYN mRNA in small-cell lung carcinomas. Serum DYN A/B levels increased in a non-small-cell lung cancer cell xenograft stress reduction mouse model. DYN B blocked cAMP formation in non-small-cell lung cancer cells. Lung tumor cells co-expressed pro-DYN and DYN, prohormone convertase 1, prohormone convertase 2, or carboxypeptidase E. DYN was observed in cancer cells infiltrating human lung tissues and nerve fibers in the bronchial submucosa. Lung cancer cells express opioid receptors and contain several combinations of opioid peptides; opioid receptor agonists blocked the growth of tumor cells. *Pro-opiomelanocortin* gene delivery blocked the growth of tumor cells and vasculature.	[208,216,217,218,219]
Myeloma	U50,488 decreased tumor cell proliferation.	[235]
Neuroblastoma	Pre-pro-DYN mRNA and pre-pro-ENK in tumor cells. Detection of opioid-binding sites in tumor cells. DYN promoted apoptosis in tumor cells; this was inhibited with isoflurane.	[216,240,242]
Pancreatic	DYN A_1-8_ did not affect tumor cell migration, chemotaxis, or invasion (MIA PaCa-2, PANC-1, BxPC-3). Mouse insulinoma beta TC3 cells: high expression of pro-DYN mRNA and derived peptides.	[186,254]
Pheochromocytoma	*Pro-DYN* gene expression and DYN release. DYN in humans. Nicotine favored the release of DYN from tumor cells. DYN A_1-17_ was the major component in pheochromocytomas, whereas DYN A_1-13_/DYN A_1-12_ were minor.	[171,257,261,262,263,264]
Prostate	DYN A, DYN A_1-13,_ and DYN A_1-7_ promote tumor cell growth. Naloxone blocked the effect mediated by DYN A and increased the growth of tumor cells at high concentrations.	[277,278]
Testicular	Pro-DYN mRNA and its derived peptides in tumor cells.	[284]

## Data Availability

Not applicable.

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
