# Peer review of "Involvement of the Opioid Peptide Family in Cancer Progression"

_biomedicines, 2023, doi:10.3390/biomedicines11071993_

Round 1
Reviewer 1 Report
the authors in this study starting from the studies that have shown how peptides mediate cancer progression by promoting mitogenesis, migration and invasion of tumor cells, promoting metastasis and anti-apoptotic mechanisms and facilitating angiogenesis / lymphangiogenesis, summarized the latest findings related to the involvement of opioid peptides (enkephalins, endorphins and dynorphins) in the development of cancer.
Do the authors think that the microbiota can also be involved? for example that helpers can read this recent manuscript.
AÄŸagündüz D, Cocozza E, Cemali Ö, Bayazıt AD, Nanì MF, Cerqua I, Morgillo F, Saygılı SK, Berni Canani R, Amero P and Capasso R (2023), Understanding the role of the gut microbiome in gastrointestinal cancer: A review. Front. Pharmacol. 14:1130562. doi: 10.3389/fphar.2023.1130562
Could the authors in the conclusions better highlight if there are clinical studies that highlight the role of peptides?
Perhaps a graphical abstract could help readers
none
Author Response
Changes in the new version are highlighted in yellow.
REVIEWER 1
Comments and Suggestions for Authors
The authors in this study starting from the studies that have shown how peptides mediate cancer progression by promoting mitogenesis, migration and invasion of tumor cells, promoting metastasis and anti-apoptotic mechanisms and facilitating angiogenesis / lymphangiogenesis, summarized the latest findings related to the involvement of opioid peptides (enkephalins, endorphins and dynorphins) in the development of cancer.
- Do the authors think that the microbiota can also be involved? for example that helpers can read this recent manuscript.
AÄŸagündüz D, Cocozza E, Cemali Ö, Bayazıt AD, Nanì MF, Cerqua I, Morgillo F, Saygılı SK, Berni Canani R, Amero P and Capasso R (2023), Understanding the role of the gut microbiome in gastrointestinal cancer: A review. Front. Pharmacol. 14:1130562. doi: 10.3389/fphar.2023.1130562
This interesting line of research has been mentioned and cited (new reference 300). See pages 44 and 63.
- Could the authors in the conclusions better highlight if there are clinical studies that highlight the role of peptides?
This has been done. See page 45.
- Perhaps a graphical abstract could help readers
This has been done.
Reviewer 2 Report
Comments:
1. Since ACTH is one of the POMC products, any ACTH effect on opioid peptide family in cancer progression?
2. In addition to glycosylation and phosphorylation, any other post-translational modifications such as acetylation, SUMOylation, methylation, etc on on opioid peptide family in cancer progression?
3. Since p53 is important for cancer progression, any p53 related info on opioid peptide family in cancer progression?
Author Response
Changes in the new version are highlighted in yellow.
REVIEWER 2
Comments and Suggestions for Authors
Comments:
- Since ACTH is one of the POMC products, any ACTH effect on opioid peptide family in cancer progression?
This has been mentioned. See page 44. A new reference (301) has been added (see page 63).
- In addition to glycosylation and phosphorylation, any other posttranslational modifications such as acetylation, SUMOylation, methylation, etc on opioid peptide family in cancer progression?
No experimental evidence to date relates the direct posttranslational changes mentioned (acetylation, SUMOylation, methylation) on opioid receptors (OR) and cancer development and progression. However, the study of indirect mechanisms of receptor desensitization and overstimulation of OR where GTPase-activating proteins (GAP), conjugated to SUMO proteins associate with MOP (For example, Rodriguez-Muñoz et al., 2007*) might be relevant and thus deserve further exploration to explore.
* Rodríguez-Muñoz M, Bermúdez D, Sánchez-Blázquez P, Garzón J. Sumoylated RGS-Rz proteins act as scaffolds for Mu-opioid receptors and G-protein complexes in mouse brain. Neuropsychopharmacology. 2007 Apr;32(4):842-50. doi: 10.1038/sj.npp.1301184.
- Since p53 is important for cancer progression, any p53 related info on opioid peptide family in cancer progression?
This has been mentioned. See page 44. Reference 302 has been added (see page 63).
Round 2
Reviewer 2 Report
No more comments